# Social observation enhances cross-environment activation of hippocampal place cell patterns

Xiang Mou[1], Daoyun Ji[1,2]*

[1]Department of Molecular and Cellular Biology, Baylor College of Medicine, Houston, United States; [2]Department of Neuroscience, Baylor College of Medicine, Houston, United States

**Abstract** Humans and animals frequently learn through observing or interacting with others. The local enhancement theory proposes that presence of social subjects in an environment facilitates other subjects' understanding of the environment. To explore the neural basis of this theory, we examined hippocampal place cells, which represent spatial information, in rats as they stayed in a small box while a demonstrator rat running on a separate, nearby linear track, and as they ran on the same track themselves. We found that place cell firing sequences during self-running on the track also appeared in the box. This cross-environment activation occurred even prior to any self-running experience on the track and was absent without a demonstrator. Our data thus suggest that social observation can facilitate the observer's spatial representation of an environment without actual self-exploration. This finding may contribute to neural mechanisms of local enhancement.

*For correspondence: dji@bcm.edu

**Competing interests:** The authors declare that no competing interests exist.

## Introduction

Social learning, defined as acquiring new knowledge through observing or directly interacting with others, is a fundamental behavior of humans and animals (*Bandura, 1997*; *Meltzoff et al., 2009*; *Heyes and Galef, 1996*). Behavioral studies have identified many potential mechanisms of how learning can occur through social observation or interaction. Besides the imitation learning that humans and some primates employ (*Bandura, 1997*; *Meltzoff et al., 2009*; *Zentall, 2006*), a common behavioral effect in many animal species is the so-called 'social facilitation' (*Zentall, 2006*; *Zentall and Levine, 1972*; *Zajonc, 1965*) or 'local enhancement' (*Heyes and Galef, 1996*): an animal's understanding of an environment is facilitated by the presence of other social subjects in the same environment. Local enhancement can be caused possibly by enhanced sensory processing due to heightened attention, acquiring environmental attributes such as safety or food availability, or other unspecified means (*Heyes and Galef, 1996*; *Zentall, 2006*; *Zajonc, 1965*). According to this idea, the presence of social subjects in an environment impacts other animals' neural processing of information related to the environment. In this study, we aim to explore the neural basis of such local enhancement effect of social observation.

In humans and rodents, spatial information of an environment is represented in spatial memory by hippocampal place cells (*O'Keefe and Dostrovsky, 1971*; *Burgess and O'Keefe, 2003*; *Wilson and McNaughton, 1993*), which fire at specific locations (place fields) of a given environment. For example, when a rat travels along a linear track from one end to the other, a subset of place cells display place fields on the track and they fire one after another in a unique sequence (*Lee and Wilson, 2002*). When the animal explores a different environment such as an open box, hippocampal place cells 'remap' (*Leutgeb et al., 2005*; *Colgin et al., 2008*; *Alme et al., 2014*), i.e.,

they either alter firing locations, stop firing, or become active. Given that spatial environments are represented by distinguished place cell activity patterns, we asked how the presence of a rat in an environment such as a track impacts an observer rat's place cell activity patterns representing the environment. Specifically, we set out to test a hypothesis that an observer's place cell sequence representing a track can be activated by another rat running on the track, even if the observer is physically in a different environment separated from the track. Through such activation of place cell activity patterns across different environments, here referred to as 'cross-activation', local enhancement may be achieved via strengthening spatial memory representation of an environment without actual self-exploration.

To test this hypothesis, we recorded neurons from the CA1 area of the hippocampus in rats as they stayed in a small box while a demonstrator rat running on a separate, nearby linear track and as they ran on the same track themselves. We found that place cell sequences during track running also appeared in the box, but only when a demonstrator was present on the track, supporting our hippocampal cross-activation hypothesis of local enhancement.

## Results

We recorded from hippocampal CA1 cells in 9 rats when they stayed in a small (25 × 25 cm) box and when they ran on a linear track. Prior to the recording, a group of 5 of these rats had gone through a training schedule that lasted 2–3 days, 15–30 min each day, where they were placed in the box while a well-trained demonstrator running back and forth for food reward on a nearby linear track (*Figure 1A*). On the first recording day, CA1 cells from these 5 rats were recorded during 3 sessions (*Figure 1B*). In two of these sessions, each of the rats stayed in the box performing the same task as in the training (Pre-box and Post-box sessions). We referred to this configuration of the recorded rat staying in the box while a well-trained demonstrator on the track as the Trained-demo box condition for a Pre- or Post-box session. In between these two sessions, the recorded rat ran the track itself for the first time (Track session). This recording procedure of 3 sessions with Pre- and Post-box under the Trained-demo condition was repeated for 1 or 2 more days. The same recording procedure continued for a total of 6–12 days, but with Pre- and Post-box under various other box conditions (*Figure 1B*), including removing the demonstrator from the track (Empty-track), removing both the track and demonstrator (No-track), replacing the well-trained demonstrator with a naïve demonstrator that had never been exposed to the track (Naïve-demo) or with a remotely controlled toy car running back and forth on the track (Toy-car), or blocking the recorded rat's view in the box while a well-trained demonstrator running on the track (Blocked-view). The other group of 4 recorded rats were each trained for 2–3 days prior to the recording, similarly as in the first group, but without the presence of a demonstrator on the track. These rats were then recorded for 6-12 days with the same Pre-box, Track and Post-box schedule each day. However, in these rats, the first 2 recording days were under the Empty-track condition, followed by the Trained-demo and other conditions. A total of 3290 putative pyramidal CA1 cells (defined as mean firing rate < 6 Hz) were recorded from all animals on all recording days. Here we analyzed 2200 cells that were active (with mean rate > 0.5 Hz) in at least one of the Pre-box, Track, and Post-box sessions on a given recording day.

### Firing sequences during rotation events in the box

In Pre- and Post-box under the conditions other than No-track and Blocked-view, a recorded rat was placed in the small box, which had opaque, high walls on its 3 sides with one side opening to the nearby linear track (*Figure 1A*). The small box and the track, both elevated from the floor, were separated by 25 cm to prevent direct physical contacts between the recorded rat and the demonstrator. The rat in the box exhibited two major types of behavior. The first was a relatively inactive behavior: facing the opening side of the box with little or small irregular movement. On average, the animals spent about 53% of the time facing the opening side of the box. However, this percentage did not differ among different box conditions (*Figure 1C*), except for the Blocked-view condition, where the opening side was blocked and rats predictably spent less percentage of time facing the opening side. This result is likely due to the fact that the opening side was the only way of exploring the outside environment regardless of the situation on the track. The second was an active, rotation behavior: the animal's body rotating clockwise (CW) or counter-clockwise (CCW), mostly starting from the

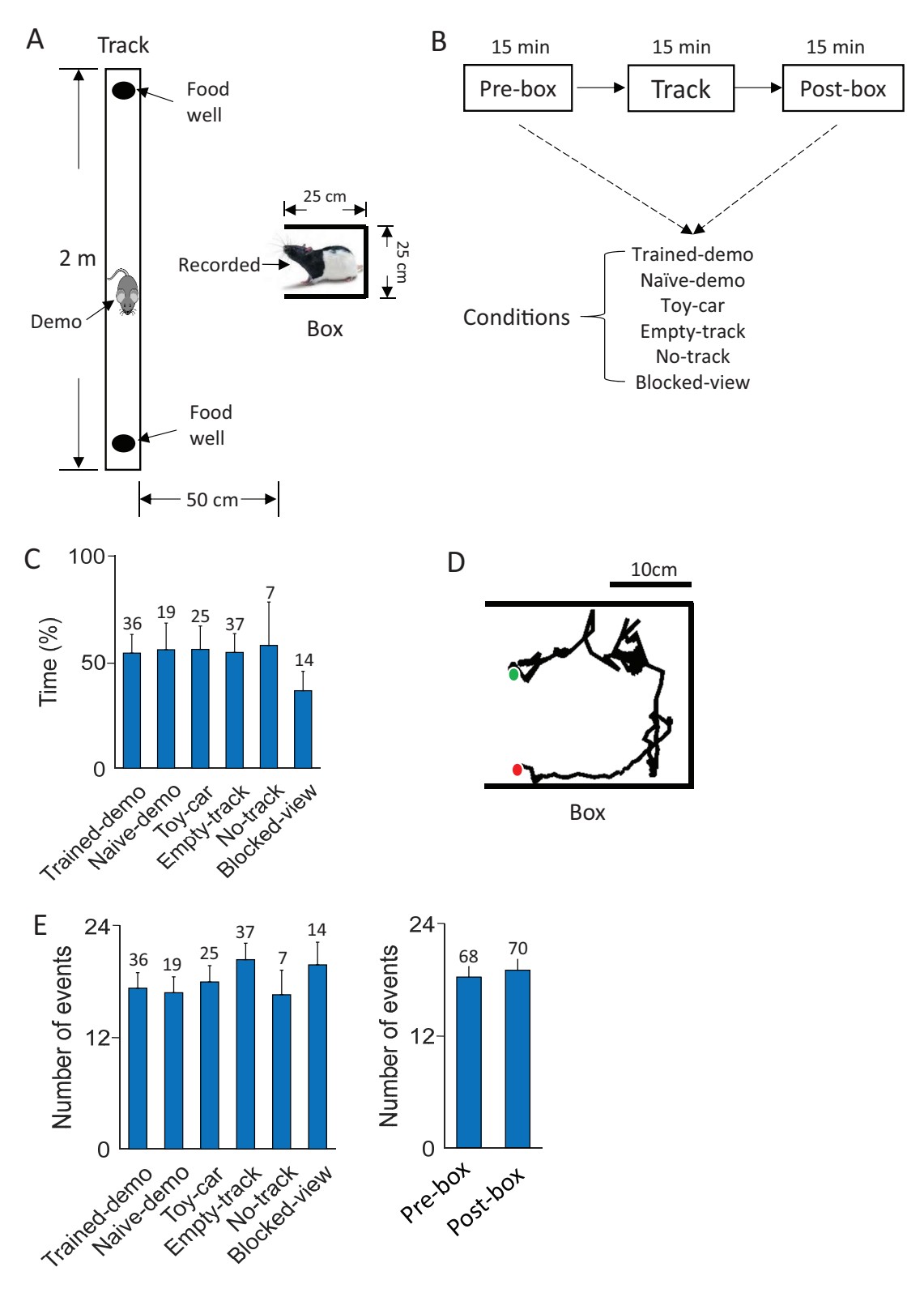

**Figure 1.** Behavioral tasks. (**A**) A recorded rat staying in a small box while a demonstrator (Demo) running on a nearby linear track. (**B**) Daily experimental procedure: staying in the box before (Pre-box) and after (Post-box) running the track (Track). The Pre- and Post-box sessions were configured with various conditions (see Results). A shows the condition with a demo (Trained-demo or Naïve-demo). (**C**) Average percentage of time animals spent facing the opening side of the box under each box condition ($F_{(5,133)}$ = 12, p=6.5 × 10$^{-10}$, comparing among all conditions one-way

*Figure 1 continued on next page*

*Figure 1 continued*

ANOVA; $F_{(4,119)}$ = 1.4, p=0.23, comparing among the conditions other than Blocked-view). Number on top of each bar: number of sessions. (D) Animal's head trajectory in an example rotation event in the box. Green/red dots: start/end positions, respectively. (E) Average number of rotation events per session under each box condition ($F_{(5,133)}$ = 1.6, p=0.17, one-way ANOVA) and in Pre-box and Post-box sessions (t = 0.46, p=0.65, t-test). Number on top of each bar: number of sessions.

The following figure supplement is available for figure 1:

**Figure supplement 1.** Rotation events were accompanied by prominent theta, not ripple, oscillations.

opening side of the box and ending at the same side (*Figure 1D*). We identified the active rotation events during all Pre- and Post-box sessions in all rats based on animals' head directions. On average, 18.7 ± 1.1 rotation events were identified per session and there was no significant difference among different box conditions or between Pre- and Post-box (*Figure 1E*). The rotation events lasted for an average of 6.67 ± 0.05 s. Analysis of CA1 local field potentials (LFPs) shows that rotation events were accompanied by strong theta (6–10 Hz) oscillations (*Buzsáki, 2002*) without much high-frequency (100–250 Hz) sharp-wave ripples (*Buzsáki et al., 1992*) (*Figure 1—figure supplement 1*). Since theta is commonly associated with active behavior such as maze-running whereas ripples mostly occur during inactive behavior such as immobility or slow-wave sleep (*Buzsáki, 2002*; *Buzsáki et al., 1992*), the analysis indicates that the hippocampus was in an active state during rotation events. In this study, we focus on the CA1 activity within these active rotation events, because our primary goal is to test our hippocampal cross-activation hypothesis by comparing CA1 activity patterns in the box and those on the track in the same type of active, theta-associated behavior. In addition, since there were multiple rotation events per session, focusing on rotation events permitted the analysis of neural activity over multiple samples of the same behavior.

We identified those CA1 cells that were active in rotation events (14.7 ± 0.4 rotation-active cells per session) and examined their firing characteristics. As illustrated by firing rate maps of CA1 cells (firing rate versus the animal's head position) in the small box (*Figure 2A*), we found that, despite the small size of the box, firing activities of many CA1 cells were not distributed across all the positions where an animal's head was located, but highly restricted. Due to the circular nature of the rotation behavior, this location specificity led to the appearance of head-direction tuning, as illustrated by a circular firing rate curve (mean firing rate versus the animal's head direction during rotation events) for each such cell in a box session (*Figure 2B*). We found that 40% of rotation-active CA1 cells (566 out of 1421) were significantly tuned to head direction during rotation events in the box (p<0.05, Rayleigh test). Consequently, simultaneously recorded CA1 cells in a box session displayed consistent firing sequences during these rotation events. When the animal rotated from different directions (CW versus CCW), the sequences appeared to be reversed to a large extent, but not always (*Figure 2C* and *Figure 2—figure supplement 1*).

To quantify the consistency of each cell's firing among rotation events of a box session, we computed a circular correlation between its firing rate curves of any two rotation events. The mean correlation among all different combinations of events in a session was compared to a distribution of correlation values obtained by random, independent shuffling of the cell's rate curve in every event (*Figure 2D*) and z-score transformed. We refer to this z-scored mean cross-event correlation as the rotation-consistency of a cell and defined cells with z-score > 1.645 (p<0.05, Z-test) as rotation-consistent. We found that 74% of rotation-active cells were rotation-consistent. On average, we identified 10.2 ± 0.5 location-consistent cells per box session. We then compared rotation-consistency of the rotation-active cells in different types of rotation events under various conditions. First, we found that the average rotation-consistency within CW or CCW events was significantly greater than that across CW and CCW rotation events of same sessions (*Figure 2E*), suggesting that the activities of these cells on average showed a dependence on rotation direction. Nevertheless, 42% of rotation-consistent cells (29% of all rotation-active cells) were also consistent between CW and CCW events. This quantification provides an explanation for many, but not always, similar firing sequences in reverse order between CW and CCW rotation events (*Figure 2C* and *Figure 2—figure supplement 1*). Second, the rotation-consistency was similar among different box conditions (*Figure 2F*). These

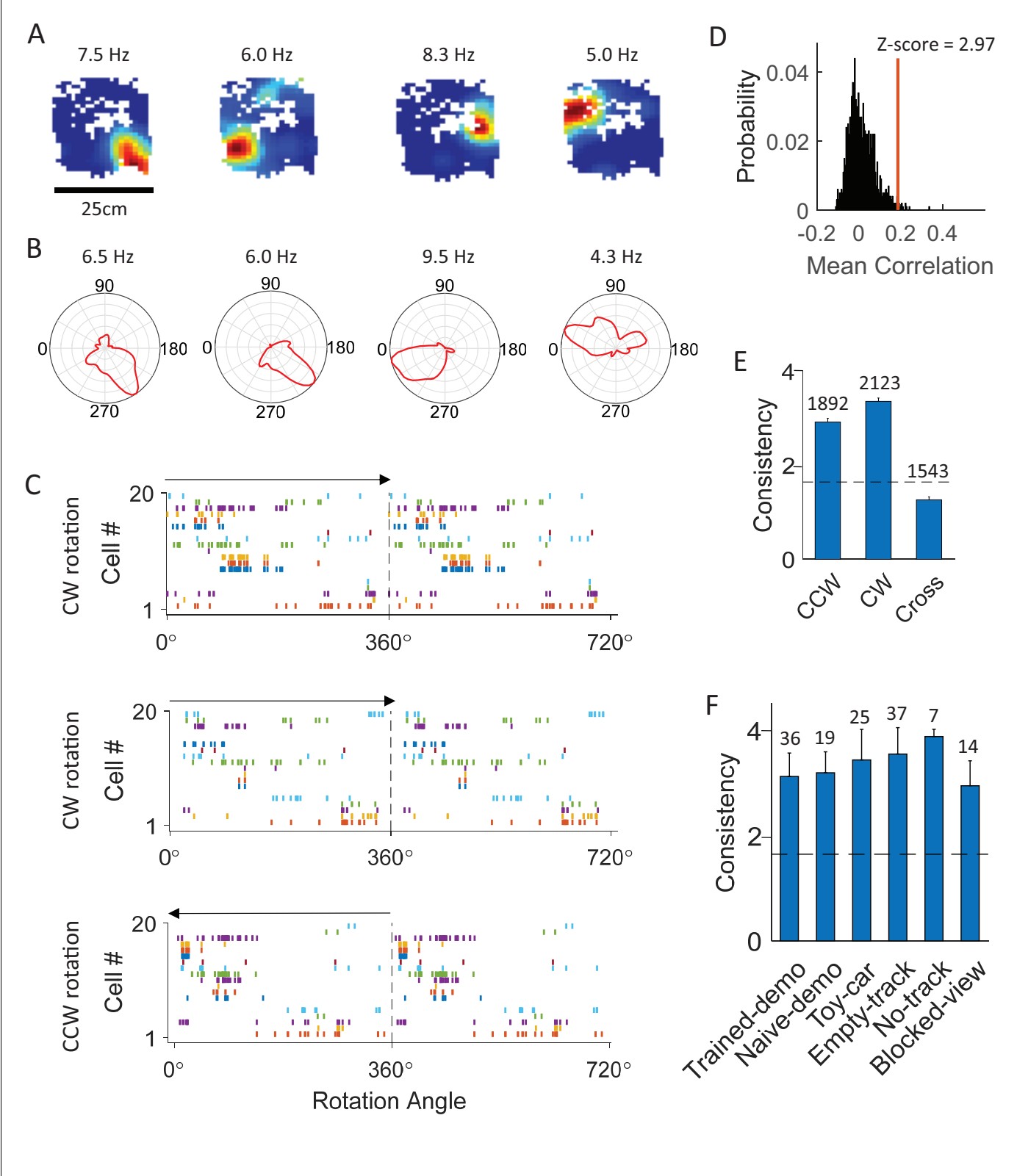

**Figure 2.** CA1 cells displayed consistent firing sequences during rotation events in the box. (**A**) Rate maps of example cells in the box during rotation events. Each color plot shows mean firing rate of a cell versus the animal's head position in the box. Number on top of each map: peak rate. (**B**) Firing rate curves of the same cells. Each plot is the mean firing rate at each bin of head direction during the rotation events. Number on the top of each curve: peak rate. (**C**) Firing sequences of all the simultaneously recorded rotation-consistent cells during a CW rotation (top), another CW rotation

*Figure 2 continued on next page*

*Figure 2 continued*

(middle), and a CCW rotation event (bottom). Each row plots spike rater of one cell versus the animal's head direction for a 360° cycle twice. Cells were ordered by their peak firing angles. Arrow: rotation direction. Note the consistent sequences during all the 3 rotation events (same firing order between the 2 CW events; reverse firing order between CCW and CW events with cells firing at similar angles during both CW and CCW events). (D) Actual mean cross-event correlation (red line) and distribution of shuffle-generated correlation values for an example rotation-consistent cell. Z-score: z-scored mean cross-event correlation (rotation-consistency). (E) Average rotation-consistency for all rotation-active cells within CW events, within CCW events and cross CW and CCW events (cross). Dashed line: threshold ($Z = 1.645$, p<0.05, z-test) for a cell to be rotation-consistent. $t = 21$, p=$3.4 \times 10^{-94}$ (paired *t*-test between cross and within-CW); $t = 17$, p=$3.5 \times 10^{-61}$ (between cross and within-CCW); Number above each bar: number of cells active in CCW or CW events, or both (only a subset active in both). (F) Average rotation-consistency under different box conditions. $F_{(5,133)} = 0.62$, p=0.62 (one-way *ANOVA* across all conditions). Each bar is the average over all the sessions (all cells within a session were averaged to get a mean value) under a condition. Number above each bar: number of sessions.

The following figure supplement is available for figure 2:

**Figure supplement 1.** Example firing sequences in a Toy-car, an Empty-track and a No-track box session.

results demonstrate that CA1 cells did not fire throughout a rotation event, but at specific locations of its rotation trajectory, and therefore formed consistent firing sequences during these events.

## Common cells between the box and the track

We showed that many CA1 cells were active during rotation events in box sessions. We then aimed to understand how their activities in the box were related to their activities during self-running on the track. To this end, we first identified individual lap events when the animal ran on the track back and forth from one end to the other (two running trajectories). We then analyzed whether rotation-active cells were also active during lap-running events on a track trajectory. On average, there were $18.8 \pm 0.9$ laps per trajectory per track session and each lap lasted for $5.8 \pm 0.2$ s. Among the 2200 cells active either during rotation in the box or during lap-running on the track across all days, 601 cells were active only during rotation, 779 cells only during lap-running, and 820 cells (common cells) during both rotation and lap running on at least one track trajectory (*Figure 3A*). These numbers indicate that, overall, 58% of all rotation-active cells and 51% of all running-active cells were common cells. However, we found that, in the Trained-demo and Naïve-demo sessions combined, these proportions of common cells (69% of all rotation-active cells, 58% of all running-active cells) were significantly higher than those in other (Empty-track, No-track, Toy-car, Blocked-view) box sessions combined (48% of all rotation-active cells, p=$3.4 \times 10^{-21}$, *binomial* test comparing with that of Trained- and Naïve-demo combined; 44% of all running-active cells, p=$7.5 \times 10^{-6}$). This finding suggests that many active CA1 cells were 'cross-activated' between the box and the track in the presence of a demonstrator, either well-trained or naïve.

We further quantified this phenomenon by computing the proportion of common cells expected from chance between each box session and a track trajectory, assuming that CA1 place cells in the box and on the trajectory were randomly and independently drawn from a common set of CA1 cells (*Alme et al., 2014*). We then compared the actual proportion with the chance proportion and defined a proportion difference index (PDI) to measure the strength of cross-activation. We found that the actual proportion was significantly higher than the chance proportion for the Trained-demo and Naïve-demo sessions, but not for others (Empty-track, No-track, Toy-car, Blocked-view) (*Figure 3B*). Similarly, the PDI was significantly greater in the Trained- and Naïve-demo sessions than other sessions (*Figure 3C*). This analysis indicates that, as long as and only when a demonstrator was present, was there cross-activation of CA1 cells significantly more than the chance level between the box and the track.

## Similar firing sequences of common cells between the box and the track

After identifying the common cells, we next asked whether firing sequences of multiple common cells were preserved between rotation events in the box and lap-running events on the track. Indeed, we found that many sequences during lap-running also occurred in rotation events, either with the same or reverse order (*Figure 4A* and *Figure 4—figure supplement 1*). To quantify this,

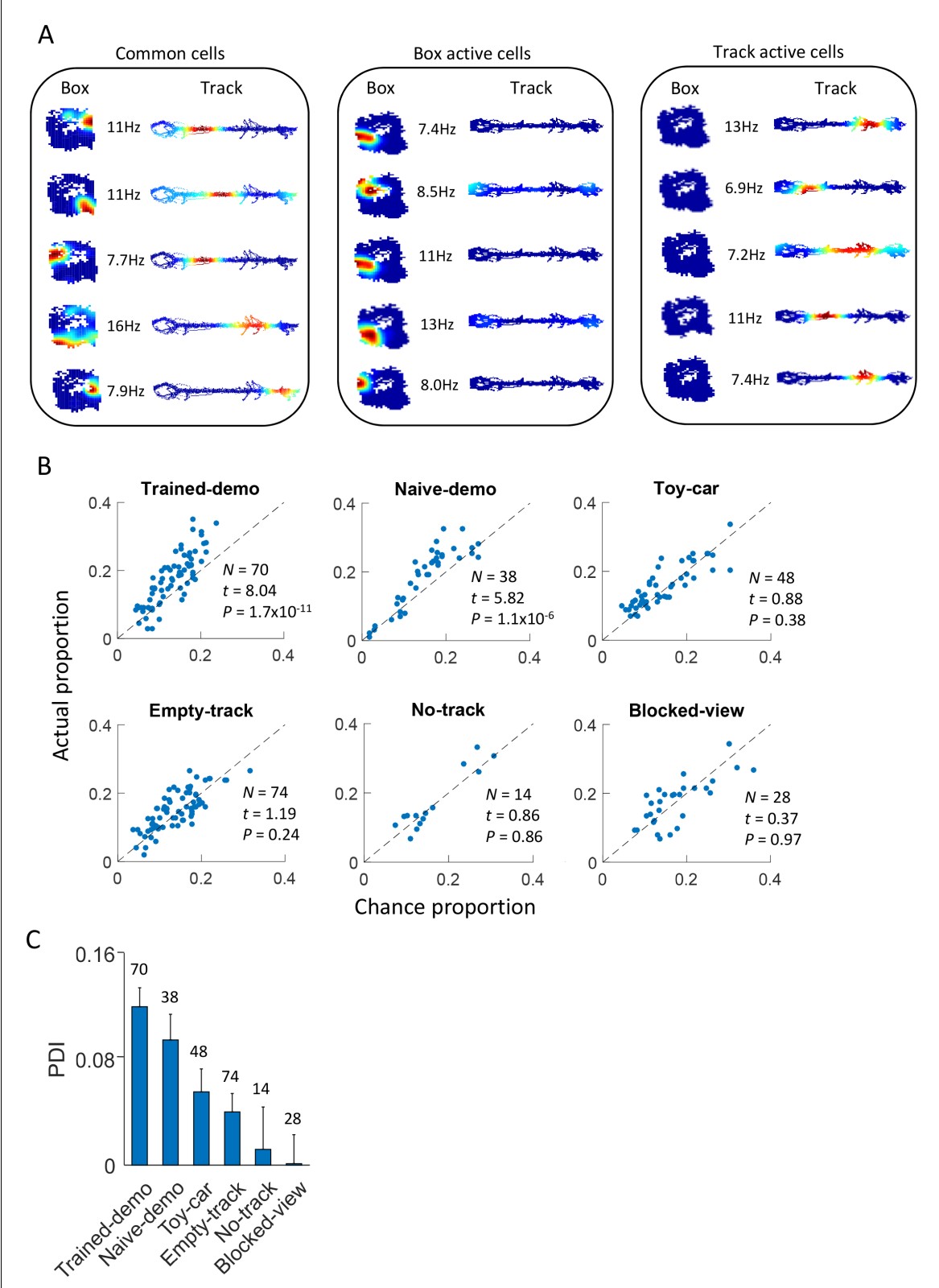

**Figure 3.** Common cells were cross-activated during rotation events in the box and during lap-running events on the track. (A) Example rate maps of common cells, those active during rotation only, and those active during lap-running only, in the same rat under the Trained-demo condition. Each row of color plots shows firing rate maps (firing rate versus position) of a cell during rotation events in a Post-box session and that of the same cell during lap-running events on a track trajectory. Numbers: peak rates. (B) Scatter plot of actual proportion versus chance proportion of common cells under

*Figure 3 continued on next page*

*Figure 3 continued*

different box conditions. Each dot represents a pairing between a box (either Pre- or Post-box) session with one of the two track trajectories on the same day (there could be up to 4 dots on each day). Dashed line: line of equal actual and chance proportion values. *N*: number of pairings between box sessions and track trajectories; *t, P*: paired *t*-test statistics and p value between actual and chance proportions. (C) Average PDI across box sessions under each observation condition. $F_{(5,267)}$ = 11, p=0, one-way *ANOVA* across all conditions. Number above each bar: number of pairings between box sessions and track trajectories.

for each recording day we constructed a template firing sequence consisting of at least 5 common cells, by ordering their peak firing locations on each trajectory of the track (at most 2 templates per day). We then obtained the sequence of these cells within each rotation event (rotation sequence) in Pre- and Post-box that contained at least 5 active common cells, by ordering their peak firing angles. We quantified the similarity between a rotation sequence and a template by a circular matching score, given the circular nature of rotation events. In our method, a high similarity score means a sequence similar to a template with the same or reverse order. We also randomly shuffled a rotation sequence 1000 times and computed the score between each shuffled sequence and the template. If a rotation sequence yielded a score with any of the templates on the same day greater than the 95 percentile of the shuffle-generated scores, we referred to it as a matching sequence.

We found that 125 out of 639 (20%) rotation sequences under Trained-demo and 73 out of 336 (22%) rotation sequences under Naïve-demo were matching sequences. In contrast, only 73 out of 712 (10%), 13 out of 116 (11%), 42 out of 419 (10%), and 22 out of 314 (7%) of rotation sequences under Empty-track, No-track, Toy-car, and Blocked-view were matching sequences, respectively. To assess how these numbers of matching sequences differed from the chance level, we generated 200 sets of randomized templates under a box condition, each by randomly shuffling cell identities in every template, and identified the number of matching sequences for each set of randomized templates, using the exactly same procedure as described. We used the proportion of randomized template sets that generated a greater number of matching sequences than our actual templates under a box condition as a measure of significance (p value). Relative to the shuffle-generated distribution of number of matching sequences, we found that the actual number of matching sequences was significant under Trained-demo and Naïve-demo, but not under Empty-track, No-track, Toy-car, or Blocked-view (*Figure 4B*). Furthermore, we also quantified the significance of matching sequences within each individual animal. In this case, we combined the rotation events under Trained-demo and Naïve-demo, in order to obtain sufficient samples for the analysis. We found a significant number of matching sequences consistently in 8 out of the 8 rats that yielded sufficient cells for the analysis (*Figure 4—figure supplement 1*). Therefore, our analysis indicates that firing sequences of common cells were preserved between lap-running and rotation events under Trained-demo and Naïve-demo conditions. Further analysis found that the precise firing times of common cells were not preserved (*Figure 4—figure supplement 2*), probably due to non-constant running speed on the track and non-constant turning speed in the box. The results nevertheless indicate that firing sequences of multiple common cells during lap-running were cross-activated in a significant number of rotation events in the box, but only when a demonstrator was present.

## Experience independence of cross-activation

Our results so far demonstrate that the CA1 activity during a rat's track running behavior was cross-activated in the box with the presence of a demonstrator on the track. A critical question is whether the track-running CA1 activity already started to appear during rotation events in the box prior to the very first experience on the track. Therefore, we analyzed CA1 cells from 5 rats in Pre-box on the first recording day under the Trained-demo condition, before they ran the track themselves for the very first time. As a key control, we also analyzed CA1 cells from other 4 rats in Pre-box on the first recording day under the Empty-track condition, prior to their first self-running on the track. We found that the actual proportion of common cells was significantly greater than the chance proportion only under Trained-demo, not under Empty-track (*Figure 5A*). Second, the actual number of matching sequences was also significantly greater than the chance level only under Trained-demo, not under Empty-track (*Figure 5B*). These results suggest that cross-activation of the track-running

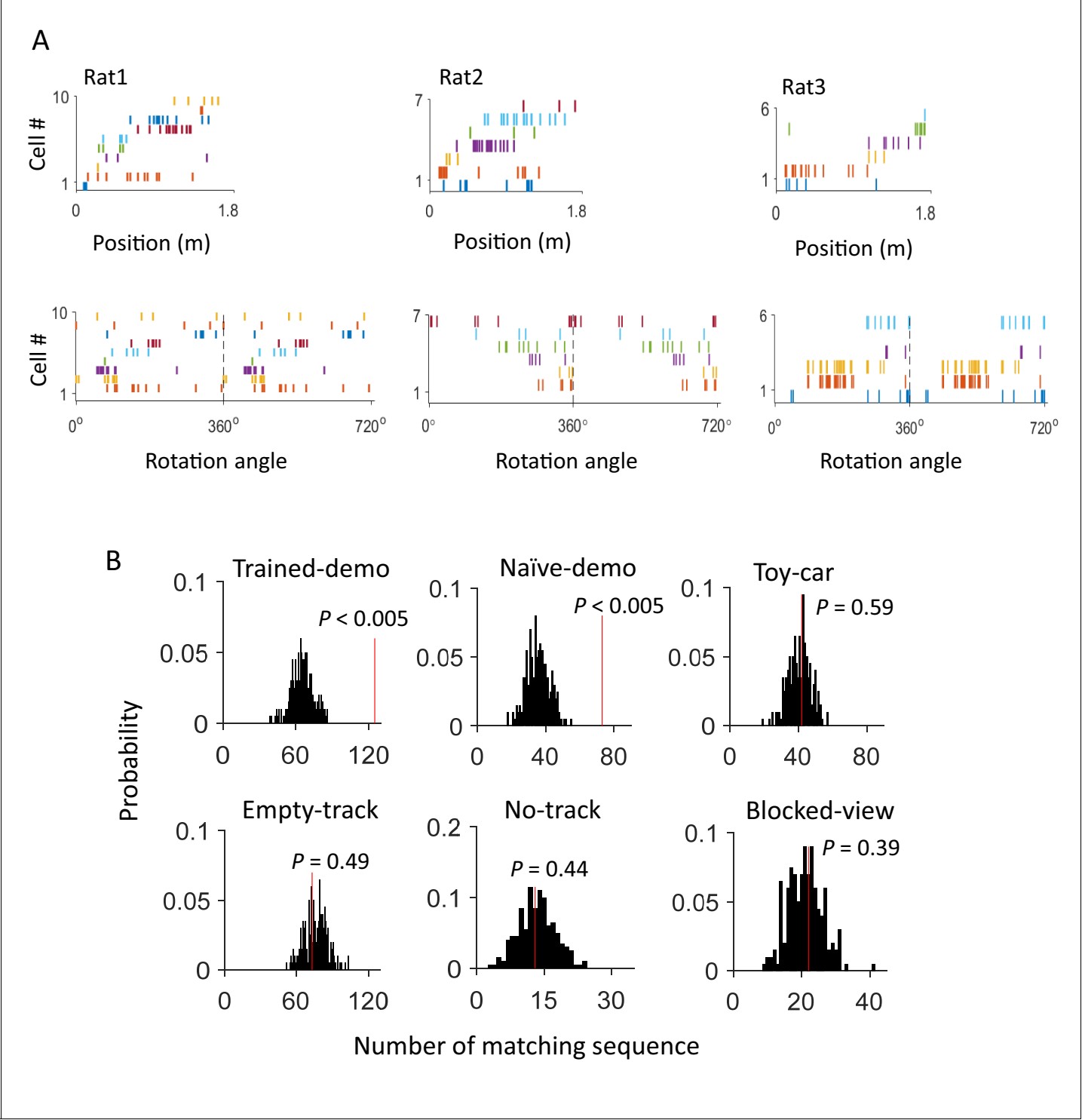

**Figure 4.** Firing sequences of multiple common cells were cross-activated between rotation events in the box and lap-running events on the track. (**A**) Firing sequences of 3 example sets of common cells (Rat1 - Rat3), each from a different rat. For each example, the sequence during a running lap on a trajectory of the track was on the top and that of the same set of cells during a rotation event in Post-box was on the bottom. Each row is the spike raster of one cell. Cells were ordered by peak firing locations on the track. A full cycles (360°) were plotted twice for each rotation event. Note that the sequences on the track matched the rotation sequence with the same (Rat1, Rat3) or opposite (Rat2) order. (**B**) Actual number of matching sequences (red line) relative to the distribution of randomly-generated number of matches (black histogram) under each box condition. *P*: significance p-values.

The following figure supplements are available for figure 4:

*Figure 4 continued on next page*

*Figure 4 continued*

**Figure supplement 1.** An example of matching sequence and the number of matching sequences in box sessions when a demonstrator was present (Demo: Trained-demo and Naïve-demo sessions combined) in each individual rat (Rat1 - Rat8).

**Figure supplement 2.** Precise firing timing among common cells did not match between matching sequences in the box and template sequences on the track trajectories.

CA1 activity was present in the box session prior to any self-running experience, but only when a demonstrator was present.

This cross-activation of place cell sequences prior to self-experience on the track raises the question of whether it was due to some kind of general, shared pre-existing activity patterns in the CA1

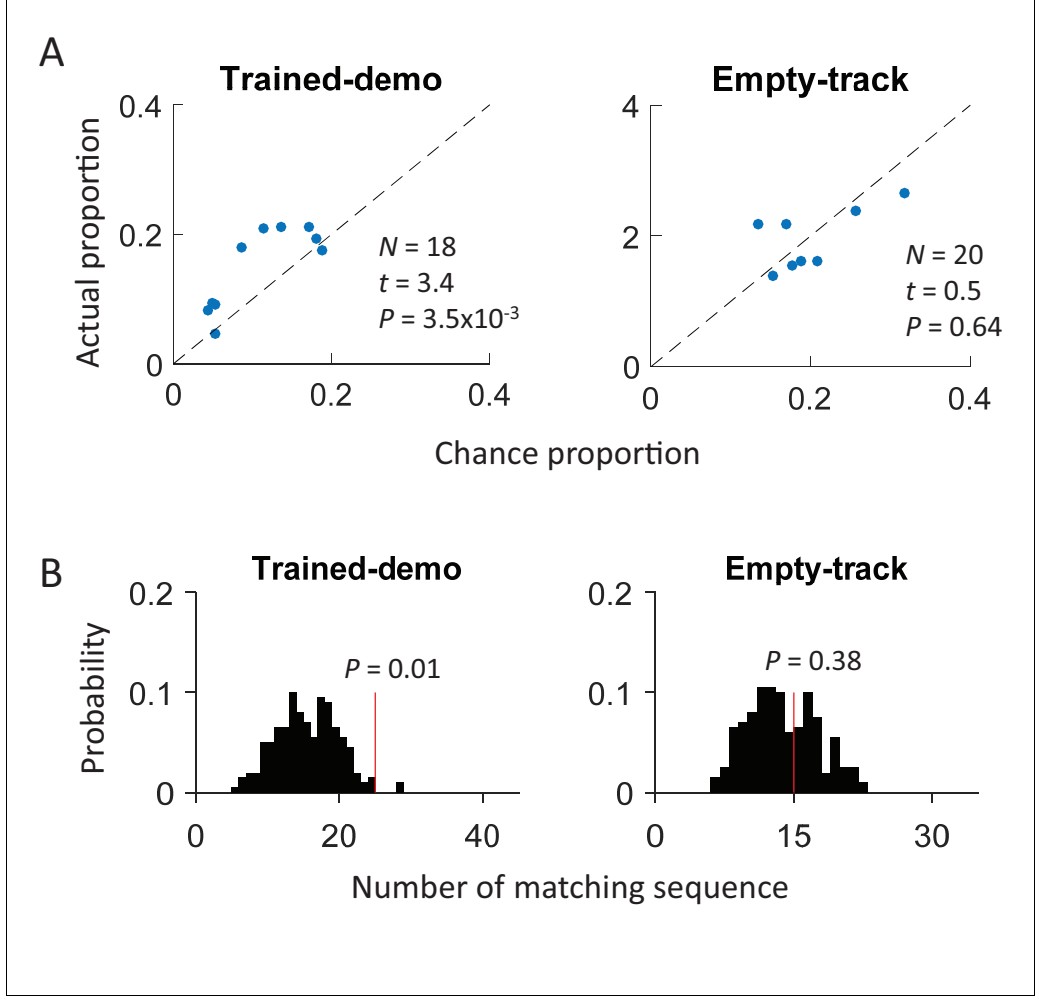

**Figure 5.** Cross-activation of CA1 activities appeared in Pre-box prior to the first self-running experience on the track. (**A**) Scatter plot of actual proportion versus chance proportion of common cells for Pre-box sessions under Trained-demo and Empty-track before the very first self-running on the track. Each dot represents a pairing between a Pre-box session and one of the two track trajectories on the same day. Dashed line: line of equal actual and chance proportion values. *N*: number of pairings between Pre-box sessions and track trajectories; *t, P*: paired *t*-test statistics and p value between actual and chance proportions. (**B**) Actual number of matching sequences (red line) relative to the distribution of randomly-generated number of matches (black histogram) in Pre-box under Trained-demo and Empty-track before first self-running on the track. *P*: significance p-values.

cell population as previous studies proposed (*Dragoi and Tonegawa, 2011*, *2013*). The fact that the cross-activation only occurred under Trained-demo or Naïve-demo, but not under other conditions, suggests that this was not the case. To further demonstrate that the cross-activation was specific to the box, we conducted additional recordings of CA1 cells from 4 rats during rest sessions before (Pre-rest) and after (Post-rest) they ran the track themselves. During these rest sessions, the animals stayed on top of a small flower pot (20 cm diameter), placed at the center of an enclosed box with 4 high walls. We detected rotation events when animals moved around on top of the flower pot with high theta activity and then identified common cells that were active both in these rotation events and in lap-running events on the track. We found that actual proportion of common cells was similar to the chance proportion (*Figure 6A*) and the actual number of matching sequences was not significantly greater than the chance level (*Figure 6B*). The analysis suggests that cross-activation was unique to the box that had an opening to the nearby track where a demonstrator was present.

Finally, although we have shown that cross-activation did not require the rat's prior self-running experience on the track, a question remains whether it did change with the self-experience. To answer this question, we examined CA1 cells on the first and second day of recording (Day1, Day2) under the same condition of either Trained-demo or Naïve-demo. We first analyzed the experience dependence at the time scale of sessions, by comparing the CA1 activities between Pre-box and Post-box of Day1 and Day2 combined. We found no significant difference in the proportion of common cells between Pre-box and Post-box (using PDI as a measure, *Figure 7A*). The actual number of matching sequences was significantly greater than the chance level in both Pre-box and Post-box (*Figure 7B*). The data thus suggest that the cross-activation in Pre- and Post-box was largely similar. We then analyzed the experience dependence at a longer time scale by comparing these same measures between Day1 and Day2 (with Pre-box and Post-box combined on each day). We found

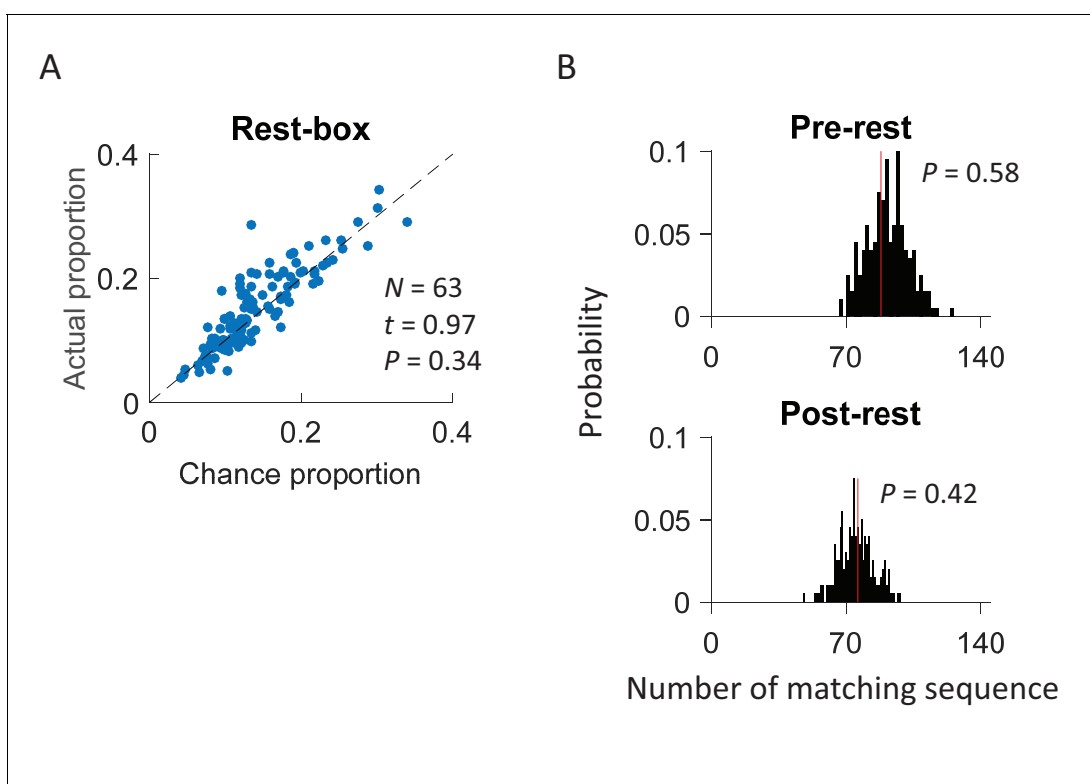

**Figure 6.** Cross-activation of CA1 activities did not occur in a rest box. (**A**) Scatter plot of actual proportion versus chance proportion of common cells for each rest session. Each dot represents a pairing between a rest session with one of the two track trajectories on the same day. Dashed line: line of equal actual and chance proportion values. *N*: number of pairings between rest sessions and track trajectories; *t, P*: paired *t*-test statistics and p value between actual and chance proportions. (**B**) Actual number of matching sequences (red line) relative to the distribution of randomly-generated number of matches (black histogram) in rotation events in the rest box. *P*: significance p-values.

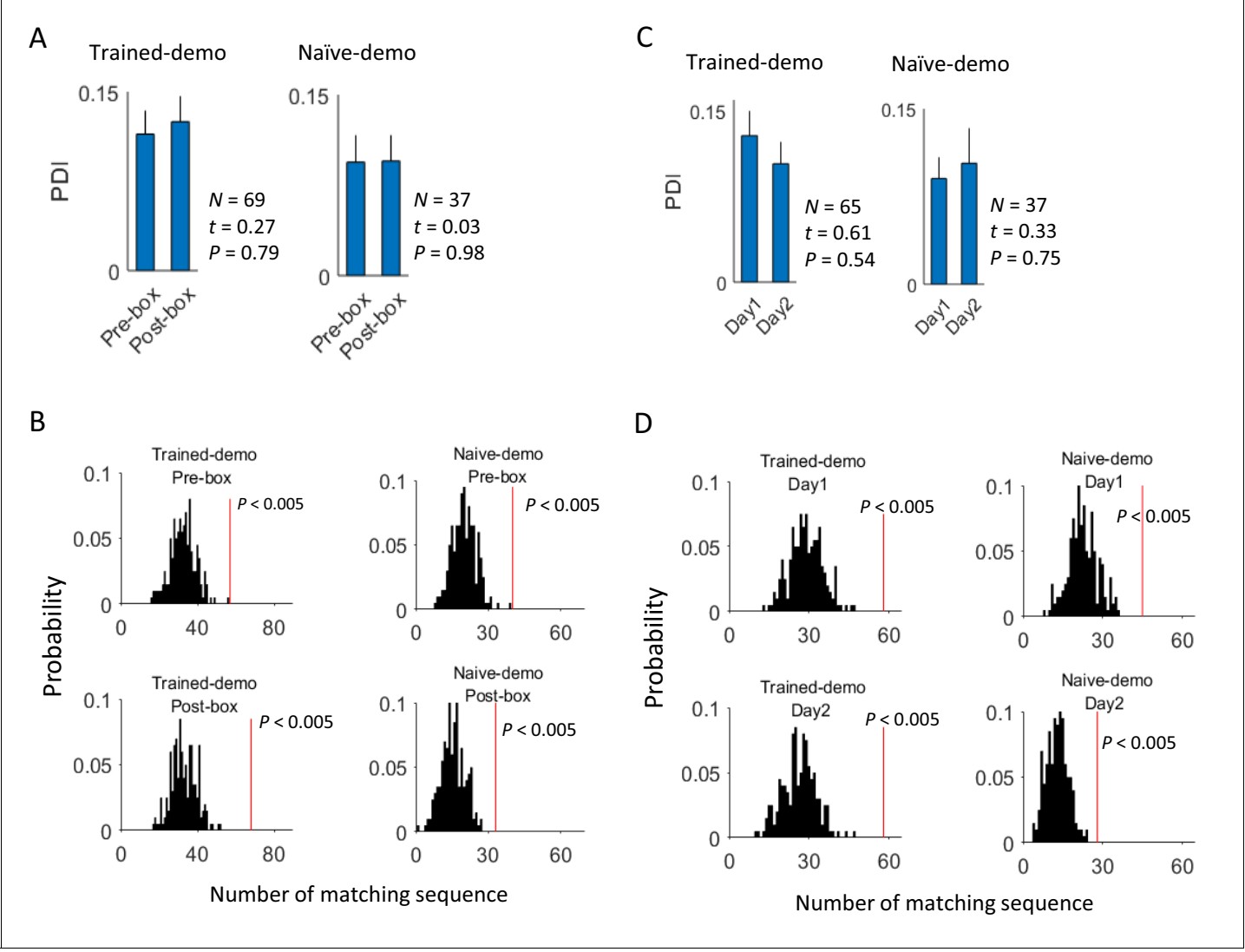

**Figure 7.** Cross-activation of CA1 activities displayed little experience-dependence. (**A**) Average PDIs of rotation-active cells for Pre-box and Post-box sessions under Trained-demo and Naïve-demo. *N*: number of pairings between Pre- or Post-box sessions and track trajectories sessions; *t, P*: paired *t*-test statistics and p value. (**B**) Actual number of matching sequences (red line) relative to the distribution of randomly-generated number of matches (black histogram) in Pre-box and Post-box under Trained-demo and Naïve-demo. *P*: significance p-values. (**C**, **D**): Same as in **A**, **B**, but for the first and second recording day (Day1, Day2) under Trained-demo and Naïve-demo.

that the cross-activation was also largely similar between Day1 and Day2 (*Figure 7C,D*). These results suggest that cross-activation of CA1 activity was insensitive to animals' self-experience on the track.

## Improved behavioral performance and place field development on the track

The finding that cross-activation already occurred in Pre-box prior to self-running on the track prompted us to explore how the cross-activation impacted the rats' behavior or their place fields on the track. We analyzed behavioral and place field parameters in the Track session on the first two days (Day1, Day2). These parameters were compared between the 5 rats that were recorded under the Trained-demo box condition (referred to as Trained-demo rats hereafter) and the 4 under the Empty-track box condition (Empty-track rats). In addition, 5 other rats were not recorded, but went through the same Pre-box, Track and Post-box procedure under either the Trained-demo (4 rats) or

the Empty-track (1 rat) condition. The behaviors of these unrecorded rats on the track were included in the behavioral analysis. We need to point out again that all these Trained-demo and Empty-track rats had gone through a training procedure in the small box for 2–3 days prior to Day1, with and without the presence of a trained demonstrator, respectively. The behavior on the track was quantified by mean running speed and number of running laps per trajectory. Both parameters in the Trained-demo rats were significantly greater than those in the Empty-track rats on both Day1 and Day2 (*Figure 8A*), suggesting an improvement in track running performance in the former group. For place field analysis, since rats were not exposed to the track before Day1, we focused on how the development of novel place fields on the track differed between the Trained-demo and Empty-track rats. Because place fields developed quickly during first a few laps (*Frank et al., 2004*) and place cells in the Trained-demo and Empty-track rats should be compared using the same number of laps, we quantified place cell properties within the first 4 laps of each trajectory on each day, by spatial information (an overall measurement of spatial specificity; *Skaggs et al., 1993*) and lap-consistency (a measure of a cell's firing reliability from lap to lap, similar to rotation consistency) (*Cheng and Ji, 2013*). We found that place cell firing on the track in the Trained-demo rats appeared to be more dispersed than those in the Empty-track rats on both Day1 and Day2 (*Figure 8B*). Indeed, spatial information of cells in the Trained-demo rats were significantly greater than those in the Empty-track rats on both Day1 and Day2 (*Figure 8C*). This improvement in spatial information was small on Day1 (15% increase in median spatial information from Empty-track to Trained-demo) and became modest on Day2 (38%). Lap consistency of cells in the Trained-demo rats also appeared greater than those in the Empty-track rats, but did not reach statistical significance (*Figure 8D*). The result suggests a mild speedup in novel place field development in the Trained-demo rats. Taken together, our data suggest that cross-activation was accompanied by an improvement in track-running performance and place field development.

## Ripple-associated replay in the box

We have focused on the rotation events in the small box, when CA1 LFPs displayed prominent theta oscillations with little high-frequency ripple oscillations (*Figure 1—figure supplement 1*). However, ripples did occur in the small box (*Figure 9A*), mostly when animals were immobile. It is known that place cell patterns on the track are often reactivated (replayed) within short ripple events of 50–400 ms during pausing/resting on the track (*Foster and Wilson, 2006*; *Diba and Buzsáki, 2007*; *Karlsson and Frank, 2009*). Here we examined whether the place cell templates on the track were also replayed within ripple events in the box during Pre-box and Post-box under various box conditions. We first identified individual ripple events in each box session. We found that the number of ripple events per unit time was slightly, but significantly increased from Pre-box (0.28 ± 0.03 ripples/s) to Post-box (0.31 ± 0.02 ripples/s) sessions ($t = 2.7$, p=0.009, paired $t$-test; $N = 72$ sessions with all conditions combined). However, the number of ripples was not significantly different across different conditions in either Pre-box ($F_{(5,66)}= 2.0$, p=0.09, *One-way ANOVA*) or Post-box ($F_{(5,66)} = 1.9$, p=0.09).

We then identified those ripple events when a template sequence on a track trajectory was replayed. Similarly to the identification of matching sequences within rotation events, we used a sequence matching method to determine whether the firing sequence within a candidate ripple event (defined with at least 5 active cells) matched a template sequence, but using a rank-correlation score instead of a circular matching score (see Materials and methods). Also similarly, we counted the number of replay events under a box condition and assessed its significance by comparing it to the distribution obtained by random shuffling of templates. First, with Pre-box and Post-box sessions combined, we found a significant number of replay events under all conditions (*Figure 9B*). The percentage of replay events among all candidate events was 7.7% under Trained-demo, 12.5% under Naïve-demo, 7.8% under Empty-track, 14.6% under No-track, 9.3% under Toy-car, and 11.2% under Blocked-view. The result suggests that ripple-associated replay occurred in the box, consistent with the previously found 'remote replay' (*Karlsson and Frank, 2009*), but the replay was comparable with or without a demonstrator. Second, we compared the replay between Pre-box and Post-box with all box conditions combined (*Figure 9C*). We found that the number of replay events was significant in Post-box (9.2%), but did not reach the statistical significance in Pre-box (6.7%), suggesting that direct track experience enhanced ripple-associated replay. Finally, we specifically analyzed the replay between the rats under Trained-demo and Empty-track on the first day's Pre-box and Post-

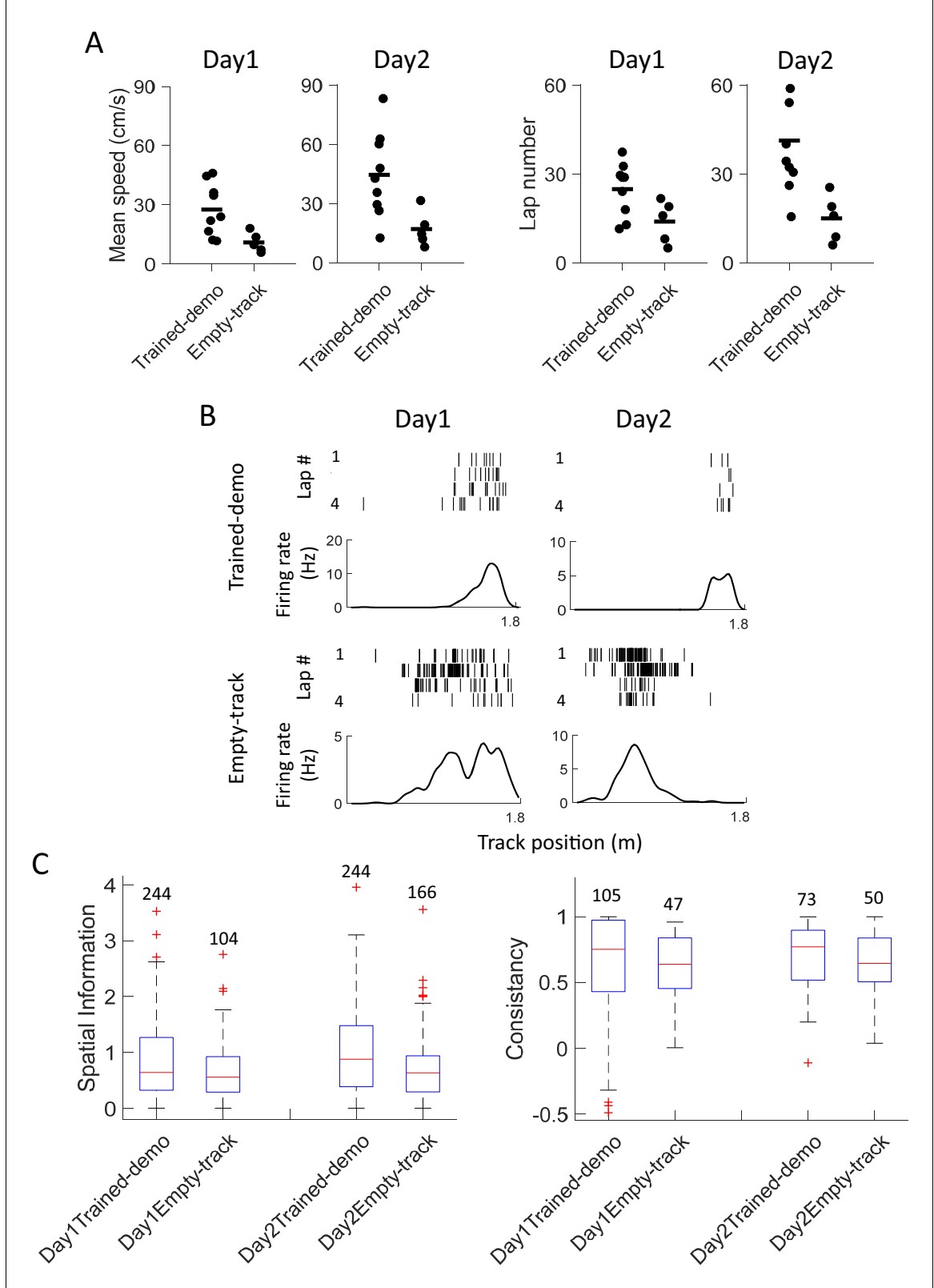

**Figure 8.** Experience in the box improved behavioral performance and novel place field development on the track. (**A**) Mean running speed and number of laps per trajectory in Track sessions of first two days (Day1, Day2) for the Trained-demo ($N = 9$) and Empty-track ($N = 5$) rats. Both parameters were significantly different between Day1 and Day2 (mean speed: $F_{1,27} = 5.3$, p=0.03; number of laps: $F_{1,27} = 4.6$, p=0.04; *Two-way ANOVA*). Trained-demo had significantly faster mean speed ($F_{1,27} = 14$, p=0.001) and ran more number of laps ($F_{1,27} = 12$, p=0.002). (**B**) Example place
*Figure 8 continued on next page*

*Figure 8 continued*

cells from a Trained-demo and an Empty-track rat on Day1 and Day2. For each condition and each day, a cell's spike raster (*top*) and its firing rate curve (*bottom*: mean rate versus position) during the first 4 laps of a trajectory are shown. Each tick is a spike. (C) Spatial information for all active cells under each condition (Trained-demo, Empty-track) on each day (Day1, Day2). Number on top of each bar: number of cells. There was a significant main effect between the conditions ($F_{1,791} = 21$, p=0, *Two-way ANOVA*), but not between days ($F_{1,791} = 1.4$, p=0.25). (D) Same as C, but for lap consistency. Although median values of lap consistent in Trained-demo appeared to be higher than those in Empty-track, there was no significant main effect between the conditions ($F_{1,274} = 0.11$, p=0.74, *Two-way ANOVA*) or between days ($F_{1,274} = 1.8$, p=0.17).

box sessions. Under both conditions, although there appeared an increase in replay from Pre-box (Trained-demo: 4.7%; Empty-track: 4.7%) to Post-box Trained-demo: 7.7%; Empty-track: 6.8%), none of the number of replay events reached the statistical significant level (*Figure 9D*), suggesting the absence of replay before the first track running experience and weak replay after. Our data are thus consistent with the idea that, in contrast to the matching sequences within rotation events, ripple-associated replay was experience-dependent and was not specific to the presence of a demonstrator.

### Lack of spatial specificity to demonstrators' positions

Besides the rotation events, animals in the small box spent much of the time facing the track (*Figure 1C*). This raises an interesting question of whether the CA1 cells in the recorded rats in the box were responding to the demonstrator's positions on the track. Raster plots of CA1 cells' spikes versus the demonstrator's track positions did not show sign of spatial tuning (*Figure 10A*). We then used spatial information and lap consistency as quantitative measurements of spatial tuning for each cell active in a box session to the demonstrator's positions. These measures were z-scored relative to those obtained using randomly shuffled spikes of the same cell (*Haggerty and Ji, 2015*). The mean z-scored spatial information and lap-consistency of all active cells under either Trained-demo or Naïve-demo were not significantly different from the chance level of 0 (*Figure 10B,C*). We found that 0.6% of cells in the Trained-demo rats and 0.4% in the Naïve-demo rats had significant spatial information ($Z > 1.645$, p<0.05), and 0% of cells in the Trained-demo rats and 0% in the Naïve-demo rats had significant lap consistency. These numbers were far below the 5% expected from chance. Therefore, our data did not support the idea that CA1 cells in the box respond to the demonstrator's positions on the track.

## Discussion

To explore the neural basis of local enhancement in social learning, we have analyzed CA1 place cells in rats as they stayed in a small box while a demonstrator running on a nearby track and as they ran the track themselves. We found that CA1 cells formed consistent firing sequences during rotation events in the box. A group of these cells (common cells) were also active during lap-running events on the track with similar firing patterns, as shown by similar multi-cell firing sequences across rotation and lap-running events. This cross-activation was specific to the box and took place only when a demonstrator (either Trained-demo or Naive-demo) was present on the track. Importantly, the cross-activation appeared before the animals' very first running experience on the track, as well as after the experience, and was accompanied by improved behavioral performance and faster place field development on the track. Our data thus show that place cell activities representing a track can be activated by staying in a separate, nearby box while another rat running on the track. This finding supports our hypothesis that presence of social subjects in an environment can enhance the activation of an observer's hippocampal place cell activity patterns that encode the environment.

We found that CA1 cells form firing sequences during rotation events in the box. Given the small size of the box, this finding is somewhat unexpected. The box size is only slightly bigger than the body length of a typical adult rat, which restricts typical walking behavior but permits body rotation. This box size is also smaller than the width of typical place fields (20–50 cm) in regular environments (*Wilson and McNaughton, 1993*). If CA1 cells in the box just fire like regular place cells with normal place fields, they would fire mostly throughout rotation events without a clear pattern. However, we found that different CA1 cells fire at different specific locations of even such a small box and multiple

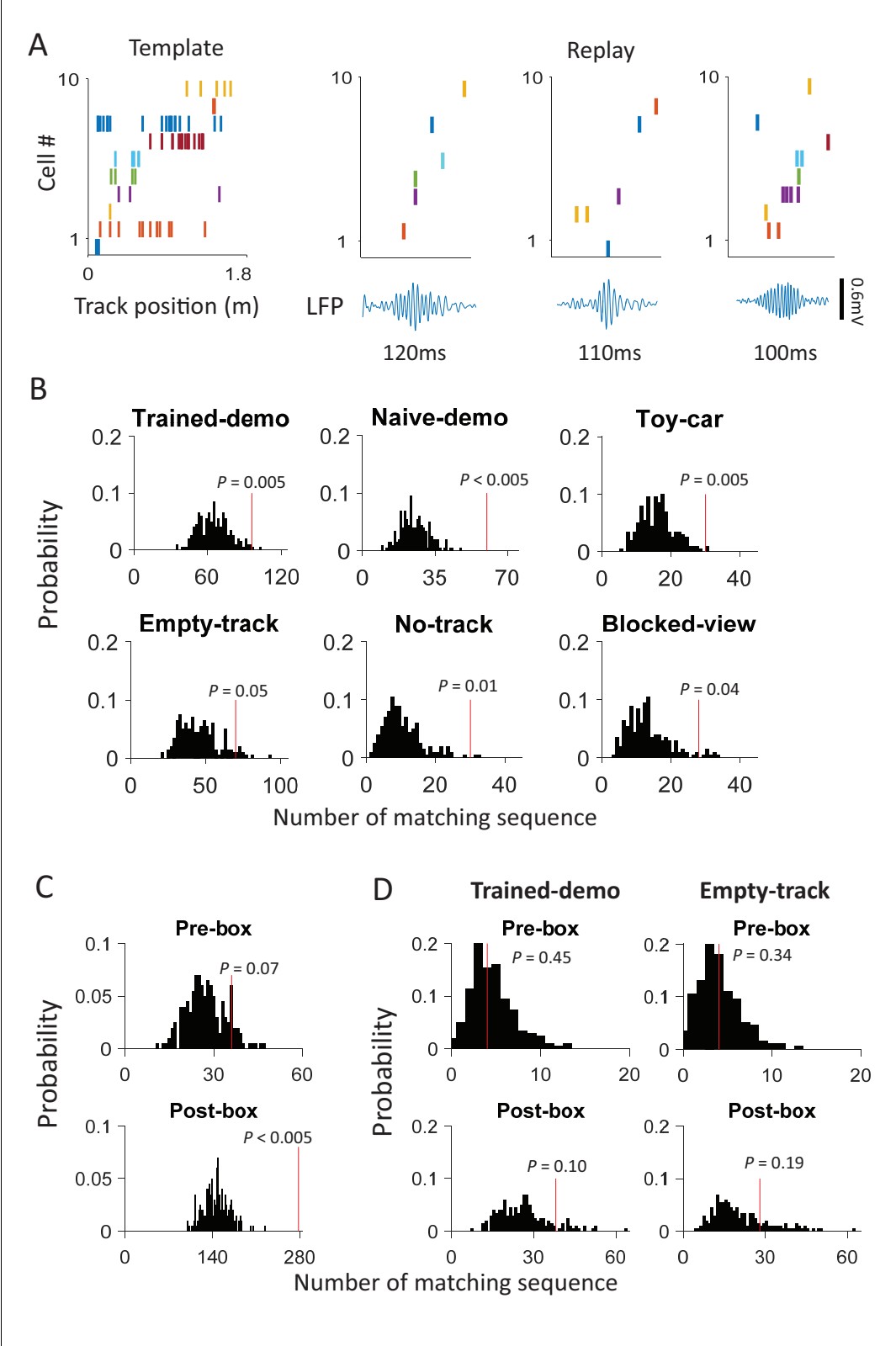

**Figure 9.** Ripple-associated replay occurred in box sessions, but was insensitive to the presence of a demonstrator on the track. (A) Example replay events in a Post-box session. *Left*: a template firing sequence made of 10 cells during a lap running event on a trajectory, plotted similarly as in *Figure 4A*. *Right*: 3 examples of replay events, each plotted with the spike raster of the same 10 cells (*top*) and the CA1 LFP (*bottom*; filtered through the ripple band of 100–250 Hz). Each tick is a spike. Note the sequential firing of the cells in each replay example, similarly as in the template. (B)

*Figure 9 continued on next page*

*Figure 9 continued*
Number of replay events (red line) relative to the distribution of randomly-generated number of replays (black histogram) under each box condition. *P*: significance p-value. (C) Same as B, but for Pre-box and Post-box sessions with all box conditions combined. (B) Same as B, but for the Pre-box and Post-box session on the very first day under the Trained-demo and Empty-track conditions.

cells fire with consistent sequences during rotation events. Then, what gives rise to these firing sequences? We used head direction tuning (circular rate curve) to quantify rotation-consistent cells during rotation events, due to the circular nature of the rotation. But these cells are unlikely true head direction cells, because they constitute a large percentage of recorded cells in our experiment, but true head direction cells, if any, are very rare in CA1 (5%) (*Leutgeb et al., 2000*). Given that rotation-consistent CA1 cells fire at consistent head positions of the rotation events and many of them fire at the same positions even when animals rotate from two opposite directions, it is possible that these cells are still place cells that are tuned to head positions, but with place fields 'squeezed' into

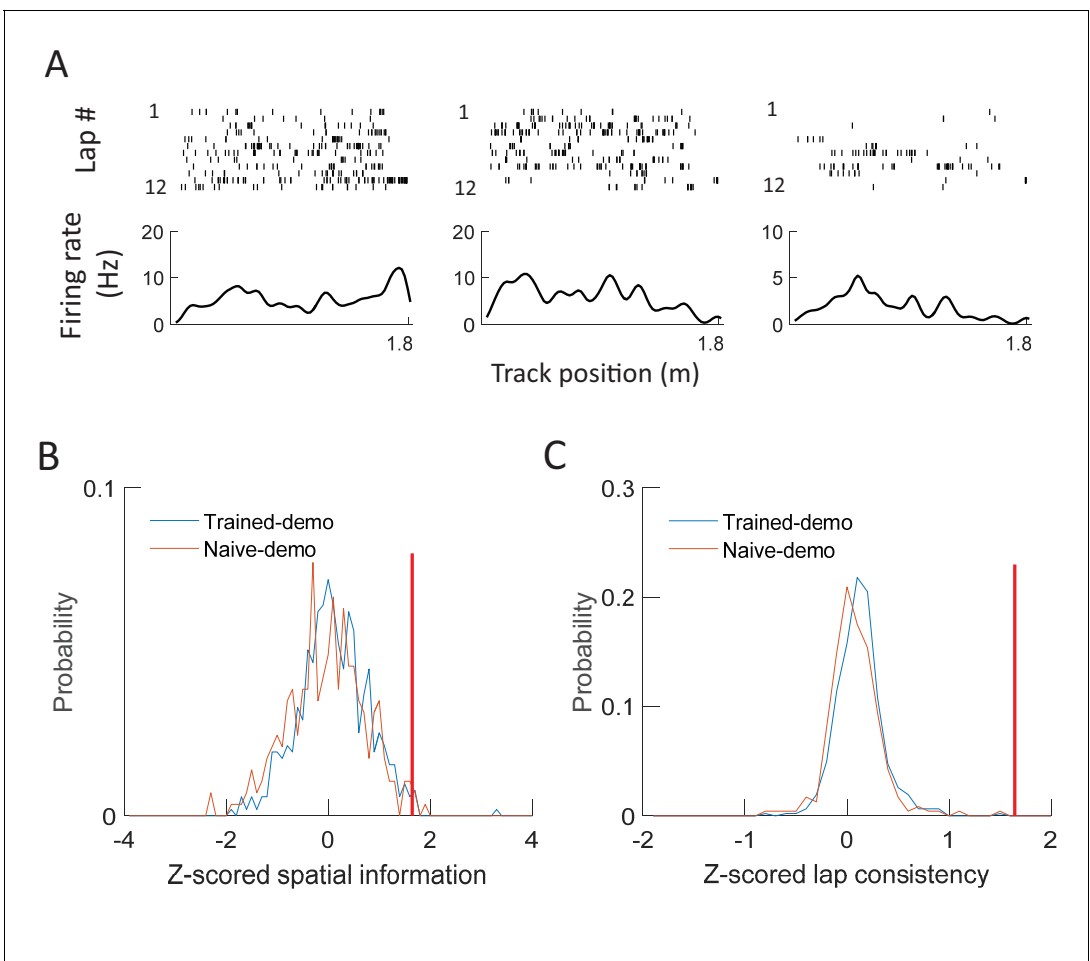

**Figure 10.** CA1 cells in the box did not respond to demonstrators' positions on the track. (A) Spikes of example cells from a rat staying in the small box, but plotted versus a well-trained demonstrator's position on the track. For each example, a cell's spike raster during every lap of the demonstrator's running on a track trajectory (*top*) and its mean firing rate curve (*bottom*: mean rate versus position) are shown. Each tick is a spike. (B) Distributions of spatial information for all active cells in all the box sessions under Trained-demo (red, *N* = 463 cells) or Naïve-demo (blue, *N* = 235). For each cell, its spatial information was z-scored relative to the spatial information values obtained by random, circular shifting of spike trains within each lap. Red vertical line: threshold to identify significant cells. The mean value of z-scored spatial information was not significantly different from 0 either under Trained-demo (*t* = 1.1, p=0.28) or Naïve-demo (*t* = −1.3, p=0.18). (C) Same as B, but for z-scored lap consistency. The mean value was not significantly different from 0 either under Trained-demo (*t* = 1.6, p=0.11) or Naïve-demo (*t* = 0.43, p=0.67).

a small box. This implies that, even if an animal is confined to a small space, place cells are tuned to specific locations within the space in active behavior and they fire with a sequence when the animal actively moves his head along a trajectory. This interpretation is consistent with the well-known phenomenon that place fields shrink as the animal's space shrinks (*Gothard et al., 1996*; *O'Keefe and Burgess, 1996*). Alternatively, the firing sequences during rotation could be internally generated to represent internal information or time, similarly as those occur when rats are running a wheel or a treadmill before resuming maze-running (*Pastalkova et al., 2008*; *Wang et al., 2015*; *MacDonald et al., 2011*, *2013*). Although we cannot rule out this possibility, the consistent head-position tuning suggests that these sequences in the box are at least strongly modulated by spatial cues.

We found that CA1 place cell activities were cross-activated between lap-running events on the track and rotation events in the box when a demonstrator (Trained-demo and Naïve-demo) was present on the same track. This cross-activation is demonstrated by a significant number of common cells between rotation and lap-running events and their matched multi-cell firing sequences between the two types of events. Under control conditions (Empty-track, No-track, Toy-car, or Blocked-view) or between the track and an enclosed rest box, the number of commons cells is similar to the chance level, suggesting independent, random remapping under these conditions, consistent with a previous report (*Alme et al., 2014*). Using the sequence matching method that utilizes high-order firing patterns of multiple common cells (*Lee and Wilson, 2002*; *Ji and Wilson, 2007*), we found a significant number of sequences during rotation events in the box that match the template sequences on the track. However, we need to point out that this cross-activation is partial, that is, only for a set of common cells, not all place cells active on the track or in the box. In fact, common cells comprise about 58% of place cells on the track under Trained-demo or Naive-demo condition and about 20% of the sequences of these cells during rotation events matched with the track templates. Since we simultaneously recorded only a small sample of CA1 cells, the actual percentage of matching sequences in CA1 is unknown, but likely higher. In any case, our data means that the overall spatial presentations of the track and the box remain relatively separated. However, a significant number of firing patterns are common in both presentations. This finding suggests the existence of common firing motifs consisting of a subset of place cells that can be activated in different environments or in different behavior (*Ravassard et al., 2013*). Importantly, our results reveal that the presence of social subjects can enhance the cross-activation of such common firing motifs of place cells across different environments.

An important question is, then, what drives the cross-activation of common firing motifs. The fact that cross-activation shows little experience-dependence suggests that it may be driven by sensory cues rather than experience. In our experimental setup, the small box had high walls on its 3 sides with the other side opening toward the track. This configuration provides common sensory input, especially distal visual cues in the room, when the animal is on the track and in the box. It is possible that this common external input biases common motifs to emerge in both environments. Interestingly, the rats' rotation events consistently started and ended at the opening side, consistent with this idea that the initiation of common firing motifs could be triggered by common external visual cues. Furthermore, under the Blocked-view condition where there was a demonstrator on the track but external room cues and the demonstrator were invisible to the rat in the small box, cross-activation did not occur, supporting a crucial role of common visual cues in cross-activation. Presumably, cross-activation could occur during any active behavior when the rat is in the small box, as long as a demonstrator is present on the track and visible. However, given that no specific task is given to the rat in the box, the animal's behavior is mostly irregular whether or not the animal is facing the opening side, which renders mostly irregular CA1 activity patterns. But the small size of the box produces one active behavior that is relatively consistent across animals and sessions: rotation behavior. According to our interpretation that the firing of CA1 cells during rotation mainly results from 'squeezed' place cell activity, the cross-activation of track templates during rotation events suggests a topological similarity between firing motifs representing the track-running trajectories and the head movement trajectories in the box. This is consistent with the proposal that place cell patterns may encode common topology in seemingly very different environments (*Dabaghian et al., 2014*).

Another important question is why cross-activation of common firing motifs occur in the box only when a demonstrator is visible on the track. One explanation is maybe related to the fact that the track is a separate environment from the box (both elevated from the floor). Without a demonstrator

on the track, the track may appear as an object to the rat in the box and thus becomes part of the visual cues that drive the place cells in the box. When there is a demonstrator moving on the track, the rat in the box may 'realize' that the track is another space within the same room that can be explored. This sense of track as an explorable space might bias the drive to a subset of place cells in both the box and the track toward shared distal room cues, and lead to the emergence of common firing motifs and thus cross-activation. However, it remains unclear whether the actual moving of the demonstrator along the track is necessary for the rat in the small box to 'realize' that the track is explorable, or merely the presence of the demonstrator on the track, e.g. staying at one end of the track, may be sufficient. Interestingly, our data show that both a well-trained and a naive demonstrator are capable of triggering cross-activation, which suggests that cross-activation does not require precise behavioral performance of the demonstrator. Also, in our data we have not found any evidence that CA1 cells of the rat in the box responded to the demonstrator's positions on the track. Therefore, the rat in the box may not necessarily understand the precise action of the demonstrator, but the presence of the demonstrator itself is sufficient to signal the spatial nature of the track (instead of merely a visual cue in the room), which leads to the cross-activation. In this way, animals can learn about an environment, without actual self-exploration. Indeed, it is known that rodents can learn a variety of tasks by observing a conspecific (*Zentall and Levine, 1972*; *Leggio et al., 2000*; *Jeon et al., 2010*), but how they do so is unknown. One theory is that, instead of imitation learning that requires detailed understanding of conspecifics' actions, rodents rely on local enhancement or social facilitation as a general scheme of social learning (*Heyes and Galef, 1996*; *Zentall, 2006*; *Zajonc, 1965*). Our finding that cross-activation can be facilitated by both trained and naïve conspecifics is consistent with the local enhancement theory and may provide a specific neural mechanism of local enhancement in social learning.

Our data also suggest possible impacts of local enhancement on animals' behavior and subsequent hippocampal activities. After 2–3 days' experience in the box with the presence of a well-trained demonstrator on the track, the animals' performance on the track is improved, compared to those animals staying in the box for 2–3 days without a demonstrator. More interestingly, their place fields during the first a few laps on the track seem already better tuned, suggesting that social observation facilitates the development of novel place fields. It is possible that this facilitation is related to the cross-activation in the box. However, a causal relationship between the two and how they are related to behavioral performance require further studies. Although ripple-associated replay may or may not take place prior to experience in other tasks (*Dragoi and Tonegawa, 2011*, *2013*; *Silva et al., 2015*), ripple replay does not occur in the box before the first self-running on the track and are similar across various box conditions in our experiment. Our data thus suggest that ripple replay in our task requires direct experience and is not impacted by social observation.

In a broader sense, this study contributes to our understanding of how hippocampal place cells are generally impacted by social behavior. Previous studies find that introducing a rat to an open box induces no change or a small modification in CA1 place cell activities, but causes remapping of cells in the neighboring CA2 area (*von Heimendahl et al., 2012*; *Alexander et al., 2016*). Whether the demonstrator causes CA1 place cells to remap in the small box was not directly investigated in our experiment. However, the fact that consistent place cell patterns occur within rotation events in the box under all box conditions suggests that the presence of a demonstrator per se would not dramatically alter place cell activities in the box, consistent with these previous studies. Instead, we found that cross-activation of common firing motifs emerges with the presence of a demonstrator on the track. This effect could be due to our task design of using a small box with one opening side, which boosts the impact of distal visual cues and the demonstrator on place cells in the box, an effort not made in previous studies. The unique influence of the demonstrator on cross-activation may come from the modification of upstream CA2 activities by social subjects. Future studies will reveal what factors and pathways are important to social influence of hippocampal place cell patterns and how they contributes to social behavior.

## Materials and methods

### Surgery and tetrode advancement

Nine Long-Evans rats (4–6 months old, male) were implanted with a hyperdrive that contained 15 independently movable tetrodes and a reference electrode, targeting the right dorsal hippocampal CA1 region (coordinates: anteroposterior −3.8 mm, mediolateral 2.4 mm relative to Bregma). Within 3–4 weeks following the surgery, tetrodes were slowly advanced to the CA1 pyramidal layer until characteristic sharp-wave ripples were observed. The reference tetrode was placed in the white matter above the CA1. Recording started only after the tetrodes had not been moved for at least 24 hours. All experimental procedures followed the guidelines by the National Institute of Health and were approved by the Institutional Animal Care and Use Committee at Baylor College of Medicine.

### Behavioral apparatuses and tasks

Our experiment setup during recording is depicted in *Figure 1A*. A small [25 cm (length) × 25 cm (width) × 40 cm (height)] plexiglass box and a 2-m long linear track were placed ~50 cm apart. Both the box and the track were elevated ~50 cm from the floor. The box had opaque, high (40 cm) walls on three sides, leaving only one side open toward the track. A separate 60 cm (length) × 60 cm (width) × 100 cm (height) enclosed rest box was placed on a table ~1 m away from the track. Animals were placed in a ceramic plate (20 cm in diameter) on top of a 30-cm tall flower pot, located at the center of the enclosed rest box. In the rest box, because animals' positions were limited in the plate, they spent time either circling around the plate or resting. The plexiglass box and the rest box were on different sides of the track. During recording experiments, the implanted rat was placed in the plexiglass box (referred to as the small box), where the rat could move freely and had full visual access to but no physical contact with the track, which was set up under several conditions (see below). When there was a demonstrator on the track, the demonstrator ran back and forth freely for milk reward at both ends. Milk reward was remotely delivered by syringe and tubing from outside a curtain separating the experimenter and recording setup. Behavioral apparatuses and tasks are described in more detail at Bio-protocol (*Mou and Ji, 2017*).

### Recording procedure

After fully recovered from surgery, the implanted rats (*N* = 9) were food deprived to 85–90% of their baseline weight. All rats first went through a pre-recording training phase: Each rat was placed in the small box for 2 or 3 days, 15–30 min each day. For a group of 5 rats in this pre-recording exposure, there was a well-trained demonstrator running on the track. The demonstrator was a male conspecific (Long-Evans rat) of similar age to the recorded rat. The well-trained demonstrator had been pre-trained on the linear track to run back and forth for milk reward for at least 1 week. For the other group of 4 implanted rats, the track was left empty without a demonstrator. Afterward, recording started and lasted for 6–12 consecutive days. On the first recording day, for the 5 rats that had seen a well-trained demonstrator in the training phase, CA1 neurons were recorded for 3 sessions (*Figure 1B*). The rats first stayed in the small box while a well-trained demonstrator (Trained-demo) running the linear track for 15 min (Pre-box session). Then, they ran the track themselves for the first time (Track session), followed by staying in the small box again while the demonstrator running on the track (Post-box session). The recorded rats had never been exposed to the track before the first recording day. On each of the following days, we set up the track in box sessions with the following different conditions: removing the demonstrator on the track (Empty-track), removing both the track and demonstrator (No-track), replacing the demonstrator with a naïve demonstrator that had never been exposed to the track (Naïve-demo) or a remotely controlled toy car (Toy-car), or staying in the box but with the view blocked while a well-trained demonstrator running on the track (Blocked-view). For the Toy-car condition, the toy car was remotely controlled by the experimenter behind a curtain. It was maneuvered to move at a speed comparable to a rat's. At the end of the track, the toy car stopped and then reversed its direction. For the Blocked-view condition, the small box was rotated 180° such that its opening side now facing a nearby wall of the room 20 cm away. Under this condition, the rat in the small box could not see either the track or the demonstrator, but had access to the auditory and olfactory information associated with the demonstrator.

Each condition was recorded for 1–3 days. For the other group of 4 rats that had seen only the empty track in the pre-recording training, the box sessions on the first day were under Empty-track, followed by Trained-demo and other conditions. In this group of rats, we also added two 30 min rest sessions (Pre- and Post-rest sessions), either before the Pre-box session and after the Post-box session or before and after the Track session, when the animals were rest on the flower pot in the enclosed rest box. Furthermore, an additional group of 5 rats were not implanted with tetrodes (therefore not recorded), but were trained and tested with the same procedure described here. Four of these rats started with a 2–3 days' training using a well-trained demonstrator, followed by the daily 3-session (Pre-box, Track, Post-box) testing schedule for 2 days under Trained-demo, as in the recorded animals. The other rat went through the similar procedure but without a demonstrator (Empty-track). The behavioral data from these unrecorded rats in the Track sessions of these two days were included in the behavioral analysis (*Figure 8A*).

## Data acquisition

Tetrode recording was made using a Digital Lynx acquisition system (Neuralynx, Bozeman, MT) as described previously (*Haggerty and Ji, 2015*). Recordings started once stable single units (spikes presumably from single neurons) were obtained. A 70 μV threshold was set for spike detection. Spike signals above this threshold were digitally filtered between 600 Hz and 9 kHz and sampled at 32 kHz. Local field potentials (LFPs) were filtered between 0.1 Hz–1 kHz and sampled at 2 kHz. The animal's positions were tracked by a red and a green diode mounted over the animal's head. Therefore, all positions presented in the rate maps were head positions. Position data were sampled at 33 Hz with a resolution of approximately 0.3 cm.

## Histology

After the recording, rats were euthanized by pentobarbital (150 mg/kg). A 30 μA current was passed for 10 s on each tetrode to generate a small lesion at each recoding site. Brain tissues were fixed in 10% formaldehyde solution overnight and sectioned at 50 μm thickness. Brain slices were stained using 0.2% Cresyl violet and cover-slipped for storage. Tetrodes locations were identified by matching the lesion sites with tetrode depths and their relative positions. All data analyzed here were recorded at the pyramidal cell layer of CA1.

## Data analysis

For each dataset (recorded from a rat on a given day), single units were sorted off-line using custom software (xclust, M. Wilson at MIT, available at GitHub repository: https://github.com/wilsonlab/mwsoft64/tree/master/src/xclust). Given that we did not track cell identities across multiple recording days, certain cells might be repeatedly sampled across days. A total of 3290 single units were obtained from 73 datasets (9 rats with 8 datasets per rat on average; a dataset included the data acquired on a recording day that comprised multiple (3) sessions). Among them, 2200 were classified as putative CA1 pyramidal cells that were active (mean firing rate between 0.5 and 6 Hz) in at least one session of the recording day. Further analyses were carried out only on these cells.

### Identification of rotation and lap-running events

We identified rotation events exhibited by rats in the small box and on top of the flower pot inside the enclosed rest box. In the small box, animals usually initiated rotation movements with their heads situated on the opening side of the box. Rotation events in the small box were defined as time periods during which the animal's head direction started on the opening side (head direction values within [−45° 45°] with the center of the opening side defined as 0°), turned either clockwise (CW) or counter-clockwise (CCW) horizontally, and terminated turning with their heads situated back on the opening side of the box (within [−45° 45°]). Rotation events with head movement spanning less than 180° were not included in the analysis. In the enclosed rest box, rotation events occurring on top of the flower plot were identified whenever an animal's head direction rotated at least 180° away from its initial location. Rotation events interrupted by irregular activities, including grooming, rearing (standing on hind legs), were excluded from further analysis. During running on the linear track, lap-running events were identified when animals ran back and forth (two trajectories) on the track. Each

time when the animal ran from one end of the track to the other on a trajectory was defined as a lap-running event.

## Common cells and chance proportion

A putative pyramidal cell was considered as active in the small box or on the linear track if its mean firing rate was between 0.5 Hz and 6 Hz in all the identified rotation events (Pre-box and Post-box combined) or in the lap-running events on at least one running trajectory. The stopping periods (>3 s of immobility) within these events were excluded from computing the firing rates. We defined common cells as those that were active both in the box and on the linear track. To quantify the significance of the proportion of common cells, we paired each box session (either Pre- or Post-box) with a running trajectory of the linear track on the same day. We then computed an 'actual proportion' of common cells active both in the rotation events in the box session and in the lap-running events on the track trajectory. Here we computed the proportions separately for the two track trajectories, because of the known fact that place cells are mostly directional on a track. Actual proportion was the proportion of common cells among all the cells active on the track trajectory. Under the assumption that the box and a track trajectory are independent environments represented by two independent, random sets of cells, the proportion of common cells by chance, chance proportion, is given by:

$$\text{Chance Proportion} = \frac{n_t}{N} \times \frac{n_b}{N},$$

where $N$ is the total number of putative CA1 pyramidal cells in a given recording day, $n_t$ is the number of active cells on the track, and $n_b$ is the number of active cells in the box. To quantify the difference between the actual proportion and chance proportion, we further defined a proportion difference index (PDI):

$$\text{Proportion Difference Index} = \frac{P_{act} - P_{chance}}{P_{act} + P_{chance}},$$

where $P_{act}$ is the actual proportion and $P_{chance}$ is the chance proportion. PDI is bounded between $[-1, 1]$.

## Rate curves, rotation-consistency, and rate maps

A rate curve was computed for each cell active on a trajectory of the track during the lap-running events on the trajectory. We divided each trajectory into 2 cm spatial bins, with the reward sites (10 cm at both ends of the track) excluded, and then counted the number of spikes occurring within each bin. The spike counts were divided by occupancy time in each bin and smoothed by a Gaussian kernel with σ of two bins to generate a rate curve. The stopping periods within these events were excluded when computing the firing rates. Similarly, a circular rate curve was computed for each cell active in the box across all rotation events. A full circle (360°) was binned into 5° arches. The firing rate of a cell in each arch was the spike count of the cell within the arch, divided by the occupancy time of the animal's head direction in the corresponding arch during rotation events. The circular rate curve was then smoothed by a Gaussian kernel with a σ of two arches. A circular rate curve was also generated for each individual rotation event. To quantify rotation consistency of a cell across rotation events, we computed a circular correlation between circular rate curves of any two individual rotation events and obtained the mean among the correlation values of all combination of individual rotation events in a box session (mean cross-event correlation). We then shuffled the circular rate cures of individual rotation events and generated 1000 sets of randomized curves. In this case, the circular rate curve of a rotation event was rotated by a random number of degrees between 0° and 360°. The circular rate curves of all rotation events in a box session were independently rotated to obtain one set of randomized curve. We computed a mean cross-event correlation value for each set of randomized curves. Rotation-consistency of the cell was the z-score of the actual mean cross-event correlation relative to these shuffle-generated correlation values. Finally, a 2-dimensional (2D) rate map was generated for a cell active during rotation events in the box. In this case, we divided the 2D space into $1 \times 1$ cm grids and computed the cell's firing rate in each bin, and then smoothed the map by a Gaussian kernel with a σ of two bins.

## Place field quantifications

We quantified parameters of place fields on the track for the first two recording days. Since we focused on novel place field development, this analysis was done only for the first 4 laps on each trajectory. For each cell active on a trajectory, we obtained its firing rate curve during the first 4 laps of the trajectory. The time periods when animals were stopping anywhere on the track during these laps were excluded from the analysis. We then computed spatial information (in bits per spike), a well-established measure of overall spatial tuning of a cell (*Skaggs et al., 1993*), from a firing rate curve. We also computed a cell's firing rate curve for each of the first 4 laps on a trajectory, we computed the Pearson correlation between a cell's rate curves of any two laps. The mean correlation averaged over all combinations of any two laps was defined as the lap consistency of the cell (*Cheng and Ji, 2013*). The lap consistency was done only on a subset of cells that were active during at least 2 of the 4 laps.

We also quantified whether a cell's firing activity in a box session under Trained-demo or Naïve-demo was tuned to the demonstrator's positions on the track. For this purpose, the position data of the demonstrator were used to identify individual laps of the demonstrator's running on each of the two trajectories on the track. For each cell active in the box session, we obtained its mean firing rate curve on a trajectory with all laps combined and then computed its spatial information from the mean firing rate curve. We also obtained a firing rate curve for each lap on a trajectory and computed lap consistency as described above. We then shuffled the cell's spike activity 1000 times, each by independent, random circular shifting of its spike train within each lap. Spatial information and lap consistency were then z-score transformed relative to the distributions of these variables obtained from the random shuffling. A cell with a z-scored spatial information or lap consistency value $\geq 1.645$ on a trajectory was considered a cell that responded to a demonstrator's position on the trajectory with significant spatial information or lap consistency.

## Template sequence construction

We constructed template sequences for identified common cells, separately for each of the two running trajectories in a Track session. To include cells with well-defined place fields in a template sequence, a cell's rate curve on a trajectory needed to have a prominent peak with peak firing rate $\geq 3$ standard deviations above its mean firing rate. Qualified cells were assigned unique identities and ordered by their peak firing locations on the trajectory to generate a trajectory template sequence, as in previous studies (*Lee and Wilson, 2002*; *Ji and Wilson, 2007*). If a rate curve had more than one peaks, the peak with highest firing rate was used. Only those template sequences consisting of at least 5 common cells were considered for the sequence matching analysis described below.

## Sequence matching

For each rotation event in the box, we examined whether the firing pattern of common cells within the event matched with any of the trajectory templates on the track on the same recording day. To do so, we first identified those commons cells in a template that were active in a rotation event. If at least 5 active common cells were identified, we computed each cell's circular firing rate curve during the rotation event and ordered them into a sequence (rotation sequence), according to the angles of their peak rates relative to 0°. We next quantified how the rotation sequence matched to a trajectory template. Due to the circular nature of firing rate curves in a rotation event, we adopted a circular-shift strategy to quantify the match. Specifically, we circularly shifted the rotation sequence by one cell at a time and computed the similarity between each shifted rotation sequence and the trajectory template by using rank-order correlation (*Lee and Wilson, 2002*; *Ji and Wilson, 2007*; *Foster and Wilson, 2006*). For example, if five common cells were ordered as 12345 as an initial rotation sequence, we generated a batch of circularly shifted sequences as 51234, 45123, and so on. The rank-order correlations could be either positive (if cells in a rotation sequence and a trajectory template had similar order) or negative (if cells in a rotation sequence and a trajectory template had opposite order). The maximum absolute rank-order correlation among the correlation values between all the shifted box sequences and the template was used as the match score.

To determine whether a match score was statistically significant, we compared it to the match scores generated from the shuffled data (*Ji and Wilson, 2007*; *Cheng and Ji, 2013*). We randomly

shuffled common cells' identities 1000 times to generate 1000 shuffled sequences. For each shuffled sequence, we carried out the same circular-shift procedure and computed a shuffled match score between the shuffled sequence and the trajectory template. If the actual match score was > 95% of the shuffled match scores, then the rotation sequence was considered as a significant match. For a rotation event, if its rotation sequence was a significant match with at least one trajectory template in the Track session on the same day, we identified it as a matching sequence.

For a group of rotation events (e.g. combining all events under Trained-demo condition or in Pre-box), we counted the total number of matching sequences that matched the corresponding templates. To determine whether the number of matches was statistically significant, we randomly shuffled each of the templates to generate 200 sets of randomized templates. Using the same procedure, we computed the number of matches for each randomized set and obtained the distribution of the number of matches generated by the shuffling. The proportion of randomized template sets that generate more matching sequences than the actual number of matching sequences was used as a significance (p) value. If the proportion was 0, the p value was denoted as p<0.005. The number of actual number of matches was considered significant if p<0.05.

## Time gap differences

To quantify whether relative firing times of cells within a matching sequence of a rotation event were similar to those during lap-running events on a trajectory, we computed time gaps of pairs of cells within each matching sequence of a rotation event and compared them to their time gaps on the corresponding trajectory. To compute the time gap on a trajectory for a pair of cells, we identified the cells' peak firing locations of their firing rate curves, computed over all laps on the trajectory (with stopping periods excluded). We computed the spatial distance between the peaks and transformed it into a time gap, based on the mean running speed on the trajectory. We then rescaled the time gaps on the track, by a scale factor that rendered the mean temporal length of lap-running events to 1. To compute the time gap of a pair of cells within a rotation event, we first scaled the event time to a total length of 1. We then found the peak firing times of the two cells' spike activities within the event and took the time interval between the peaks as the time gap of the pair in the rotation event. We subtracted the two time gaps and averaged it over all pairs within a matching sequence to obtain a time gap difference for each matching sequence. We then shuffled the time intervals of cells with a matching sequences 200 times, each by randomly shifting a cell's spike train with all cells' firing order maintained and the total length of the rotation event then scaled back to 1. The actual time gap difference was z-scored relative to the distribution of all the time gap differences obtained using the shuffled spike trains. The distribution of all z-scores of all the matching sequences under a condition (Trained-demo, Naïve-demo) was plotted.

## Ripple event identification and ripple replay

We first filtered CA1 LFPs in a box session within the ripple band (100–250 Hz). Individual ripple events were identified by a peak threshold of 4 standard deviations (stds) of the filtered LFPs. Start and end times of ripple events were detected using a threshold of 2.5 stds. Those ripple events with at least 5 common cells active were considered candidate events. For each candidate event, the peak firing times of all active cells were identified and arranged into a sequence (ripple sequence). We then quantified whether the ripple sequence matched a template sequence on the track, similarly as in detecting matching sequences of rotation events. However, since ripple sequences were not circular in nature, we used the rank-order correlation between a ripple sequence and a template as a score of sequence similarity, instead of the circular matching score. The significance of a score was similarly evaluated by random shuffles of ripple sequences and templates.

## Statistics

For parametric statistics, data are expressed as mean ± SEM. Multiple group comparisons were conducted using ANOVA. Two group comparisons were done using Student's *t*-test or paired *t*-test. p<0.05 was considered significant. For randomized permutation test, p values were defined as the proportion of randomized samples larger than the actual values. p=0 means that a p value was too small to report.

## Additional information

### Funding

| Funder | Grant reference number | Author |
| --- | --- | --- |
| National Institute of Mental Health | R01MH106552 | Daoyun Ji |
| Simons Foundation | 273886 | Daoyun Ji |

The funders had no role in study design, data collection and interpretation, or the decision to submit the work for publication.

### Author contributions

XM, Conception and design, Acquisition of data, Analysis and interpretation of data, Drafting or revising the article; DJ, Conception and design, Analysis and interpretation of data, Drafting or revising the article

### Author ORCIDs

Xiang Mou, http://orcid.org/0000-0002-8579-7316
Daoyun Ji, http://orcid.org/0000-0003-4115-5888

### Ethics

Animal experimentation: All experimental procedures in this study followed the guidelines by the National Institute of Health and were approved by the Institutional Animal Care and Use Committee at Baylor College of Medicine (protocol #AN-5134).

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
