## [Decision Letter]

Thank you for submitting your article "Cross-activation of hippocampal place cell patterns by social subjects" for consideration by *eLife*. Your article has been reviewed by three peer reviewers, and the evaluation has been overseen by a Reviewing Editor and Timothy Behrens as the Senior Editor. The reviewers have opted to remain anonymous.

The reviewers have discussed the reviews with one another and the Reviewing Editor has drafted this decision to help you prepare a revised submission.

All of the reviewers were enthusiastic about the findings, which highlight a role for the hippocampus in processing social information. In general, all of the reviewers also judged the study well designed, the analyses appropriate, and the results robust. However, each reviewer had several questions and recommendations for improvement that should be incorporated into a revision. Overall, there was a general concern that the paper is not written in a way that is very accessible to the general readership, and is possibly only somewhat understandable to neurobiologists without expertise in hippocampus. Therefore, the authors should make every effort to clearly explain the findings and their importance without jargon in their revised manuscript.

Other specific concerns that must be addressed are as follows:

1) In the Introduction, the authors write, "When the animal explores a different environment […] another group of place cells becomes active". This is not entirely accurate. CA1 place cells have previously been reported to express place fields in different places (Leutgeb et al., Science 2005) and to express multiple place fields in large environments (Park et al., PLoS ONE 2011). So, active CA1 place cell populations would be expected to overlap to a certain extent for two environments placed in the same room (i.e., the conditions of the present study).

2) In the Results, the authors need to clarify the following passage: "despite the small size of the box, firing activities of many CA1 cells were not distributed across all the positions where an animal's head was located, but highly restricted (Figure 2)". Where is the animal's head positioned in these rate maps?

3) In the Results, the authors write, "This analysis indicates that, as long as and only when a demonstrator was present, was there a cross-activation of a group of CA1 cells between rotation in the box and lap-running on the track". This is stated too strongly. There was still some cross-activation when no demonstrator rat was present, according to the reported cell proportions.

4) In the Results, the authors write, "by comparing the CA1 activities between Pre-box and Post-box". Was this for Days 1 and 2 combined?

5) The authors should perhaps cite and discuss Alexander et al., Nature Communications 2016, which presents results relevant to the present study.

6) The third sentence of the subsection “Recording procedure” does not make sense and should be revised.

7) The authors should provide degrees of freedom when reporting statistics (e.g., "F = 9.2" in the Figure 3 legend).

8) Throughout the text, it was unclear to me what exactly the reported n-values represent. For example, in the subsection “Data analysis”, the authors refer to "73 datasets". What exactly is a dataset? A collection of running sessions? What exactly do the n's presented in the scatterplots (e.g., Figure 3) represent? The legend states "number of sessions", but this is still unclear. Does this represent the number of box sessions across all rats for each condition? What does the n = 63 represent exactly in Figure 6? The legend states n = number of pairs, but this was still confusing to me. Were there 63 rest sessions across all rats? How do these numbers relate to the 73 datasets mentioned in the Methods? These are just a few examples; this issue persists throughout the Methods and Results.

9) In the subsection “Identification of rotation and lap-running events”, the authors need to clarify what is meant in the passage ending with "since the rest box was enclosed". Wasn't the rest box a flower pot? How is that enclosed?

10).In the subsection “Common cells and chance proportion”, the authors refer to "firing rate". Was this the mean firing rate or in-field firing rate?

11) In the subsection “Common cells and chance proportion”, the authors state that a cell was considered active if its firing rate was between 0.5 and 6 Hz. Why did the authors put an upper limit on firing rate? To exclude fast-spiking cells (i.e., putative interneurons)?

12) In the subsection “Rate curves, rotation-consistency, and rate maps”, the authors mention random shifting. How exactly were the data shifted?

13) In the subsection “Template sequence construction”, the authors state that, to be included in a template sequence, a cell had to have a peak firing rate >= 3 SD above its mean firing rate. Is there a reference for this method of template selection (e.g., is this how template sequences were selected in reference 10?) or did the authors implement this method for the first time?

14) In the Figure 1—figure supplement 1 legend, the authors state that no peak in the ripple frequency range was observed in the power spectrum. But, ripples are not stationary, whereas theta is. So, would one expect to see a peak for ripples using this method of power estimation? Did the authors see a peak in the ripple frequency range during immobility?

15) In several of the figures, the authors state that spike rasters are plotted against two rotation cycles, implying that spikes occurred across two successive cycles. Instead, it appears as though the authors simply plotted the same data twice.

16) From which condition are the examples in Figure 3? (With or without a demonstrator rat? Pre-box or Post-box?) How were these examples selected? Are they from the same rats or different rats?

17) The third cell in Figure 3 in the "track active cells" group is also active in the box.

18) The authors focus on the order of neuronal discharge when assessing the similarities between the box and track sequences. They ignore however the more precise temporal relationship that exists across the active cell assembly. In this experiment both the box and track sequences occur on the same 'clock'- the theta rhythm. Thus if the box sequences are being used to code for space on the track it would be expected that not just the mere order, but also the temporal aspects of the assembly would be preserved. This could be examined various ways, one possible approach would be to apply Bayesian decoding, based on the decoded positions on the track, to estimate trajectories encode during the circling-related discharges in the box (similar to Silva et al., Nature Neuroscience, Dec 2015). This would allow for the classification of the events as true trajectory events (based on high correlation/low error) and quantitative assessment of the similarity of the trajectory sequences based on parameters such as maximum jump distance and absolute correlation (see Silva et al., Nature Neuro, Dec 2015).

19) The authors suggest the presence of the 2nd rat running on the track allows the observer to 'realize' the track is another space. What they do not control for however is the mere presence of the second animal. More specifically, if the observer could see the 2nd rat sitting next to the track or even running on a separate distinct track, would the results be the same? This is key to understanding if the sequences are specific to the track the demonstrator rat is exploring (context specificity), as well the possibility that any social cue alters the strength or duration of the putative plasticity linking the sequence of active neurons in the box, thus altering the probability of the neurons maintaining some order in the subsequent context. Please comment on this issue.

20) The authors hypothesize that the common cells are driven by the sensory information overlapping between the box and track sessions- this seems easy to test experimentally by altering cues between the two sessions. Please comment on this issue.

21) Evidence based on the reactivation of these same sequences during ripple events would greatly strengthen the argument that there is link between the observation and the 'learning' of the environment. Have the authors looked for ripple-associated reactivation? Are there differences in the number of ripples or the level of reactivation between the different experimental conditions (demo/no track/toy/etc.)?

---

## [Author Response]

*All of the reviewers were enthusiastic about the findings, which highlight a role for the hippocampus in processing social information. In general, all of the reviewers also judged the study well designed, the analyses appropriate, and the results robust. However, each reviewer had several questions and recommendations for improvement that should be incorporated into a revision. Overall, there was a general concern that the paper is not written in a way that is very accessible to the general readership, and is possibly only somewhat understandable to neurobiologists without expertise in hippocampus. Therefore, the authors should make every effort to clearly explain the findings and their importance without jargon in their revised manuscript.*

We have tried our best to revise the wording and tone in the manuscript to make it more accessible to non-experts.

*Other specific concerns that must be addressed are as follows:*

*1) In the Introduction, the authors write, "When the animal explores a different environment […] another group of place cells becomes active". This is not entirely accurate. CA1 place cells have previously been reported to express place fields in different places (Leutgeb et al., Science 2005) and to express multiple place fields in large environments (Park et al., PLoS ONE 2011). So, active CA1 place cell populations would be expected to overlap to a certain extent for two environments placed in the same room (i.e., the conditions of the present study).*

We agree with the reviewers on the point that spatial environments are encoded by place cell ensembles and there are overlapping place cells in two different environments. Therefore, we have revised the sentence to “… they either alter firing locations, stop firing, or become active”.

2) In the Results, the authors need to clarify the following passage: "despite the small size of the box, firing activities of many CA1 cells were not distributed across all the positions where an animal's head was located, but highly restricted (Figure 2A)". Where is the animal's head positioned in these maps?

In all the rate maps, firing rates were (color-) plotted versus positions in the small box or on the linear track. The positions were monitored by two diodes on the animal’s head. Therefore, all positions were head positions. We have clarified this in this passage and in the Materials and methods section.

*3) In the Results, the authors write, "This analysis indicates that, as long as and only when a demonstrator was present, was there a cross-activation of a group of CA1 cells between rotation in the box and lap-running on the track". This is stated too strongly. There was still some cross-activation when no demonstrator rat was present, according to the reported cell proportions.*

Part of the sentence has been rephrased to “… was there cross-activation of CA1 cells significantly more than the chance level between the box and the track”. Here we point out that our data show that the overlap under the control conditions (without a demonstrator) was random. That is, cells active in the two environments are randomly drawn from the same pool of cells. This is consistent with a study (Alme et al., PNAS, 111:18428, 2014) that is specifically designed to examine this matter.

*4) In the Results, the authors write, "by comparing the CA1 activities between Pre-box and Post-box". Was this for Days 1 and 2 combined?*

The data (shown in Figure 7) was for Day1 and 2 combined. This has been clarified in this paragraph.

*5) The authors should perhaps cite and discuss Alexander et al., Nature Communications 2016, which presents results relevant to the present study.*

We have now cited and discussed this study, as well as another relevant one suggested by another reviewer.

*6) The third sentence of the subsection “Recording procedure” does not make sense and should be revised.*

Part of the text has been rephrased as “CA1 neurons were recorded for 3 sessions (Figure 1). The rats first stayed in the small box while a well-trained demonstrator (Trained-demo) running on the linear track for 15 minutes (Pre-box session). Then, they ran the track themselves for the first time (Track session), followed by staying in the small box again while the demonstrator running on the track (Post-box session).”

*7) The authors should provide degrees of freedom when reporting statistics (e.g., "F = 9.2" in the Figure 3 legend).*

Degrees of freedoms have been added in figure legends.

*8) Throughout the text, it was unclear to me what exactly the reported n-values represent. For example, in the subsection “Data analysis”, the authors refer to "73 datasets". What exactly is a dataset? A collection of running sessions? What exactly do the n's presented in the scatterplots (e.g., Figure 3) represent? The legend states "number of sessions", but this is still unclear. Does this represent the number of box sessions across all rats for each condition? What does the n = 63 represent exactly in Figure 6? The legend states n = number of pairs, but this was still confusing to me. Were there 63 rest sessions across all rats? How do these numbers relate to the 73 datasets mentioned in the Methods? These are just a few examples; this issue persists throughout the Methods and Results.*

We thank the reviewer for pointing this out. We have now made every effort to add more details and clarify the meaning of each N, and how it was derived from our datasets either in the figure legend, in the Results section or in the Materials and methods section. We believe the clarity on this matter has been greatly improved. Regarding the reviewer’s specific questions here, “73 datasets” refers to 73 recording days (from a total of 9 rats, 6 – 12 days per rat). Each dataset includes multiple recording sessions (e.g. the Pre-box, Track, Post-box sessions) on the same day. In Figure 3, we quantified the fraction of cells active in each box session (Pre-box or Post-box) that were also active when the animal ran one of the track trajectories on the same day (2 track trajectories per day). Each dot was one such pairing of a box session and a trach trajectory. There could be up to 4 pairings on a given day (a dataset). N is the total number of pairings combined from all animals for a given condition. The plot in Figure 6 is similar to Figure 3, except for that the box sessions were replaced by rest sessions. N = 63 means that there were 63 pairings of rest sessions and track trajectories.

*9) In the subsection “Identification of rotation and lap-running events”, the authors need to clarify what is meant in the passage ending with "since the rest box was enclosed". Wasn't the rest box a flower pot? How is that enclosed?*

We have now revised this sentence, as well as the paragraphs under “Behavioral apparatuses and tasks”, in the Materials and methods section to better describe the rest box configuration. The rat was sitting on a plate on top of a flower pot. The flower pot was placed in the middle of an enclosed box.

*10) In the subsection “Common cells and chance proportion”, the authors refer to "firing rate". Was this the mean firing rate or in-field firing rate?*

We meant mean firing rate. We have clarified this.

*11) In the subsection “Common cells and chance proportion”, the authors state that a cell was considered active if its firing rate was between 0.5 and 6 Hz. Why did the authors put an upper limit on firing rate? To exclude fast-spiking cells (i.e., putative interneurons)?*

Yes, to exclude putative interneurons. We only analyzed putative active pyramidal cells (rate between 0.5 and 6 Hz). We have now clarified this in this paragraph and other places in the Materials and methods section as well.

*12) In the subsection “Rate curves, rotation-consistency, and rate maps”, the authors mention random shifting. How exactly were the data shifted?*

We have revised this sentence in the Materials and methods section to clarify this issue. To shift a circular rate curve, one cell’s firing rates in binned 5⁰ arches of a full circle (360⁰) were rotated by a random number of degrees between 0⁰ and 360⁰. This random rotation was independently repeated for each individual rotation event. When we checked our data for this response, we realized that we had accidentally presented the Pearson correlation instead of circular correlation data in Figure 2. We have corrected this mistake in the figure and the corresponding text. The conclusion remains the same.

*13) In the subsection “Template sequence construction”, the authors state that, to be included in a template sequence, a cell had to have a peak firing rate >= 3 SD above its mean firing rate. Is there a reference for this method of template selection (e.g., is this how template sequences were selected in reference 10?) or did the authors implement this method for the first time?*

We implemented this threshold to include “true” place cells with prominent place fields. We are not aware of other studies using this specific threshold. However, this is equivalent to other published methods of thresholding, either using number of spikes, place field parameters, or spatial information. For example, Lee and Wilson, 2002 uses number of spikes per lap as a threshold. In any case, a small number of “non-spatially tuned” cells should not be included in templates.

*14) In the Figure 1—figure supplement 1 legend, the authors state that no peak in the ripple frequency range was observed in the power spectrum. But, ripples are not stationary, whereas theta is. So, would one expect to see a peak for ripples using this method of power estimation? Did the authors see a peak in the ripple frequency range during immobility?*

The reviewer is right that ripples are intermittent, individual events, but they occur frequently during immobility and slow-wave sleep. When the power is averaged across a long period of time of same immobility or slow-wave sleep, a prominent peak around 140-180 Hz frequently emerges. Here the plots are for the rotation events in the box only. The absence of a peak in the ripple band suggests that ripples only rarely occur within rotation events, if any, which is the purpose of this plot. Ripples did occur outside the rotation events in the box. We have detected individual ripple events and analyzed the number of ripples and associated replay in a new figure (Figure 9).

*15) In several of the figures, the authors state that spike rasters are plotted against two rotation cycles, implying that spikes occurred across two successive cycles. Instead, it appears as though the authors simply plotted the same data twice.*

The review is right that we simply plotted the same data twice. Because 0⁰ and 360⁰ corresponded to the same location in the box, the spikes at the beginning and end of the cycle were actually responding to the same location. Therefore, we believe plotting firing patterns twice illustrates the location responses better. We have revised the figure legend to clarify this.

*16) From which condition are the examples in Figure 3? (With or without a demonstrator rat? Pre-box or Post-box?) How were these examples selected? Are they from the same rats or different rats?*

Those cells were from the same rat, in the same Post-box session with the presence of a well-trained demonstrator. We have clarified this in the figure legend.

*17) The third cell in Figure 3 in the "track active cells" group is also active in the box.*

We thank the reviewer for pointing this out. We have replaced this cell with a different one.

*18) The authors focus on the order of neuronal discharge when assessing the similarities between the box and track sequences. They ignore however the more precise temporal relationship that exists across the active cell assembly. In this experiment both the box and track sequences occur on the same 'clock'- the theta rhythm. Thus if the box sequences are being used to code for space on the track it would be expected that not just the mere order, but also the temporal aspects of the assembly would be preserved. This could be examined various ways, one possible approach would be to apply Bayesian decoding, based on the decoded positions on the track, to estimate trajectories encode during the circling-related discharges in the box (similar to Silva et al., Nature Neuroscience, Dec 2015). This would allow for the classification of the events as true trajectory events (based on high correlation/low error) and quantitative assessment of the similarity of the trajectory sequences based on parameters such as maximum jump distance and absolute correlation (see Silva et al., Nature Neuro, Dec 2015).*

We thank the reviewer for raising this important question of whether the precise firing timing of common cells, not just their firing order, was similar between the box and the track. We have performed new analysis to address this question. Because the circular nature of the firing sequences in the box complicates the Bayesian detection of trajectory events, we opted to directly compute firing time gaps of place cells with neighboring place fields on the track. We quantified the difference between these gaps on the track and their time gaps within firing sequences of rotation events in the box. This actual difference was then compared with what would be expected if the firing times of cells within rotation events were randomly shifted but with firing orders maintained. The result (Figure 4—figure supplement 2) shows that the actual differences were not significantly smaller than the differences obtained by randomly shifted sequences. We believe this is an expected result, because moving speed in both the lap-running events on the track and the rotation events in the box were not constant. As a result, even though lap running and rotation events use the same theta clock, the number of theta cycles between any two cells were different.

*19) The authors suggest the presence of the 2nd rat running on the track allows the observer to 'realize' the track is another space. What they do not control for however is the mere presence of the second animal. More specifically, if the observer could see the 2nd rat sitting next to the track or even running on a separate distinct track, would the results be the same? This is key to understanding if the sequences are specific to the track the demonstrator rat is exploring (context specificity), as well the possibility that any social cue alters the strength or duration of the putative plasticity linking the sequence of active neurons in the box, thus altering the probability of the neurons maintaining some order in the subsequent context. Please comment on this issue.*

We agree with the reviewer that the presence of a rat may be sufficient for the observer to “realize” that the track is another space. We have added this possibility in the Discussion section. We realize that many interesting manipulations in our behavioral task can be performed in future to explore what factors contribute to the social effect, for example, the precise action of the demonstrator as the reviewer mentioned, the spatial arrangement between the track and box, or the social relationship between demonstrator and observer.

*20) The authors hypothesize that the common cells are driven by the sensory information overlapping between the box and track sessions- this seems easy to test experimentally by altering cues between the two sessions. Please comment on this issue.*

We performed the experiment with an additional condition: the recorded rat was in the small box while a well-trained demonstrator was running the track, but the small box was blocked so that the recorded rat could not see the track, as well as other room cues. We have now analyzed the data and found little cross-activation under this condition, as in other control conditions. This result is consistent with the idea that sensory information is important for cross-activation. We have incorporated the data under this condition to Figure 1–Figure 4 with some of the numbers modified. We have also included this result in the Discussion.

*21) Evidence based on the reactivation of these same sequences during ripple events would greatly strengthen the argument that there is link between the observation and the 'learning' of the environment. Have the authors looked for ripple-associated reactivation? Are there differences in the number of ripples or the level of reactivation between the different experimental conditions (demo/no track/toy/etc.)?*

We have now added the analysis on the ripple-associated sequence replay in Figure 9. Our result indicates that replay did occur in box sessions. Further analysis suggest that the replay was increased from Pre-box to Post-box, but did not occur significantly on the first day of track running and did not differ across conditions. Our data thus suggest that ripple replay mainly depends on direct running experience on the track and may not be involved in the effects of social observation.